# Mapping the Quantitative Dose–Response Relationships Between Nutrients and Health Outcomes to Inform Food Risk–Benefit Assessment

**DOI:** 10.3390/foods14081420

**Published:** 2025-04-20

**Authors:** Gabriel Mateus, Cíntia Ferreira-Pêgo, Ricardo Assunção

**Affiliations:** 1Egas Moniz Center for Interdisciplinary Research (CiiEM), Egas Moniz School of Health & Science, Caparica, 2829-511 Almada, Portugal; 111305@alunos.egasmoniz.edu.pt; 2Centro Cardiovascular da Universidade de Lisboa, CCUL (CCUL@RISE), Centro Académico de Medicina de Lisboa, CAML, Faculdade de Medicina, Universidade de Lisboa, 1649-028 Lisbon, Portugal; cintia.pego@ulusofona.pt; 3CBIOS—Universidade Lusófona’s Research Center for Biosciences and Health Technologies, Av. Campo Grande 376, 1749-024 Lisbon, Portugal; 4Food and Nutrition Department, National Institute of Health Doutor Ricardo Jorge (INSA, IP), Av. Padre Cruz, 1649-016 Lisbon, Portugal

**Keywords:** nutrient dose–response, health outcomes, risk–benefit assessment, dietary modifications, novel foods

## Abstract

In the context of the global food system transformation towards sustainable and healthy diets, risk–benefit assessment supported by quantitative dose–response relationships between nutrients and health outcomes is crucial for evaluating the public health impact of dietary modifications, particularly those involving novel foods. The primary aim of this review was to establish a foundational basis for RBA by compiling and synthesising quantitative dose–response relationships identified through a comprehensive literature review. This review encompassed the last 15 years up to March 2025, utilising databases such as PubMed, Scopus, and Web of Science. This review prioritised recent meta-analyses of observational studies and randomised controlled trials with a low risk of bias, as assessed using the ROBIS tool. This review identified significant dose–response relationships across 12 nutrients and their associations with various health outcomes. While dietary fibre has demonstrated broad protective effects, cereal fibre has been found to be the most beneficial for colorectal cancer prevention. Calcium has been shown to have inverse associations with several cancers, although high dairy intake may increase the risk of prostate cancer. Haem iron was linked to an increased risk of several chronic diseases, whereas non-haem iron showed less consistent associations. Zinc exhibited a potential U-shaped relationship with colorectal cancer risk. These findings underscore the complexity of nutrient–health relationships, highlighting the importance of considering dose–response gradients and nutrient sources. Although this overview primarily summarises quantitative associations without exploring in-depth mechanistic or population-specific details, it underscores the complexity of nutrient effects, including nonlinearity and source dependency. This analysis offers a valuable foundation for future risk–benefit assessments of various food scenarios, thereby informing dietary recommendations and public health strategies.

## 1. Introduction

The global imperative for sustainable and healthy diets is driving significant transformations in food systems, prompting the exploration of novel food sources and dietary patterns [1,2,3,4]. Amidst this transition, evaluating the public health impact of dietary modifications, particularly those involving novel foods, requires a robust assessment methodology [5]. To address the need for more comprehensive and robust evaluations, risk–benefit assessment (RBA) offers a more thorough approach than traditional risk assessment, which primarily focuses on hazards. RBA simultaneously considers both the potential risks and benefits of dietary choices, thereby providing a holistic evaluation [3,6,7,8,9,10]. Therefore, the primary aim of RBA is to comprehensively assess the overall health impact of food consumption, thereby informing dietary policies and guidelines [6], particularly when considering dietary modifications involving novel food components, such as edible insects or dietary substitutes for conventional foods [5].

Expanding upon conventional risk assessment methodologies, RBA adapts a four-step risk assessment process to encompass the nutritional, microbiological, and toxicological outcomes of food consumption [3]. The aforementioned steps comprise the following: (1) establishing the framework and scope of the analysis, encompassing problem definition and scenario identification; (2) selecting and prioritising relevant health effects for evaluation; (3) quantifying risks and benefits through individual assessments and overall health impact estimations; and (4) comparing scenarios to interpret the results and guide effective communication [3]. A fundamental aspect of the RBA methodology is the rigorous establishment of nutrient dose–response relationships, specifically the quantitative association between dietary intake and health outcomes [3,7]. This emphasis on quantitative dose–response data is crucial for predictive modelling, the accurate assessment of dietary modification impacts, and guided study selection [3,11]. Quantitative characterisation through dose–response analysis is an essential preliminary step within standard RBA frameworks, facilitating the meaningful integration or comparison of risks and benefits [3,6,8]. The resulting body of evidence establishes a basis for assessing the risk–benefit profiles of various dietary scenarios, such as replacing red meat with novel alternatives. This addresses a critical need within the context of evolving dietary guidelines and sustainable food systems [1,2,3]. Nevertheless, the selection of components for inclusion in RBA presents certain challenges, including the inherent difficulty in obtaining sufficient evidence for all potentially relevant components and the necessity for transparent, systematic methodologies, such as those proposed by Boué et al. [6], to guide this selection process [6,12].

Given the continuously evolving body of evidence on the relationships between nutrients and health outcomes, it is imperative to utilise current evidence to inform rigorous RBAs. Therefore, the primary aim of this study was to comprehensively identify and compile published quantitative dose–response relationships between predetermined nutrient intakes and specific health outcomes within the RBA framework. The primary emphasis is on the extraction and summarisation of quantitative metrics, such as relative risks or hazard ratios per intake increment, as reported in recent high-quality meta-analyses. This approach is preferred over a comprehensive examination of the underlying biological mechanisms, absorption kinetics, or detailed population-specific variations for each nutrient, which would require a different scope of review. Accordingly, the present review aims to (1) comprehensively identify and evaluate quantitative associations between the dietary intake of selected nutrients and various health outcomes; (2) provide a comprehensive evidence base for assessing the risk–benefit profiles of diverse food scenarios, particularly in the context of novel food sources and dietary modifications; and (3) contribute to the development of robust assessment methodologies for evaluating the public health impact of dietary modifications, particularly those involving novel foods.

This review was conducted to address the following research questions:What are the established dose–response relationships between selected nutrients and various health outcomes, as documented in the current scientific literature?What are the direction and magnitude of these dose–response relationships, specifically whether they are protective or harmful and to what extent?Are there complexities identified within these dose–response relationships, such as nonlinear patterns, threshold effects, or variations contingent upon the nutrient source or dietary context?What are the principal data gaps and areas requiring further investigation to enhance the understanding of nutrient dose–response relationships for risk–benefit assessment?

## 2. Materials and Methods

This study employed a comprehensive review methodology, using structured literature searches, to synthesise evidence on nutrient dose–response relationships.

### 2.1. Search Strategy

A comprehensive literature search was performed using the PubMed, Scopus, and Web of Science databases to identify relevant meta-analyses reporting quantitative dose–response relationships for each nutrient under investigation. Studies published within the past 15 years (up to March 2025) were included. A combination of keywords representing essential concepts (such as nutrient names, dietary intake, dose–response relationships, and meta-analyses focusing on quantitative effect estimates) were used, linked by Boolean operators (AND, OR), to identify the pertinent literature in each database. The search strategy and results are presented in Appendix A. We also manually searched the reference lists of the retrieved articles to avoid missing relevant papers.

### 2.2. Inclusion Criteria

Studies were selected based on the following inclusion criteria: (1) meta-analyses of observational studies and randomised controlled trials (RCTs); (2) evaluation of dose–response relationships for dietary nutrients and health outcomes; (3) prioritisation of studies with quantitative dose–response data, encompassing effect estimates (e.g., relative risks, hazard ratios) and confidence intervals across diverse intake levels; and (4) prioritisation of studies assessed as having a low risk of bias using the ROBIS tool [13]. In cases where multiple meta-analyses satisfied these criteria for the same nutrient–outcome pair, preference was given to those with the lowest risk of bias. This was followed by the consideration of the most recent publication date and the clarity of the quantitative dose–response data presented. The nutrients under evaluation were predetermined and selected from the “final list” and “short list” identified in the methodological framework developed by Boué et al. [6]. Although this framework was initially developed and exemplified in the context of substituting red meat with novel food sources such as *Acheta domesticus* (house crickets), the nutrient dose–response relationships examined in our review are not limited to this specific scenario and can inform risk–benefit assessments across a variety of dietary changes.

### 2.3. Exclusion Criteria

Studies were excluded if they (1) exhibited a high risk of bias based on the ROBIS assessment, except in instances where no low-bias alternatives were available for a specific nutrient–outcome pair; (2) did not provide quantitative dose–response data; or (3) focused on nutrient supplementation rather than dietary intake. In situations where multiple meta-analyses were available for a nutrient–outcome pair, preference was given to those with a lower risk of bias and/or more recent publication dates.

### 2.4. Study Selection

Initial electronic database searches identified 313 studies in PubMed, 837 in Scopus, and 797 in Web of Science, resulting in a total of 1947 records. Following a preliminary screening of titles and abstracts and the application of inclusion and exclusion criteria, 160 articles were retained. After removing duplicates, this number was further reduced to 92 articles. From this collection, meta-analyses were selected according to predefined inclusion and exclusion criteria, culminating in the final selection of 50 articles corresponding to 60 nutrient–health outcome pairs across 12 nutrients (Figure 1). The final selection for each nutrient–health outcome pair predominantly comprised meta-analyses that integrated evidence from observational studies, primarily prospective cohort and case–control studies, with a lesser emphasis on randomised controlled trials and cross-sectional studies.

### 2.5. Data Extraction

Data extraction was performed to capture dose–response relationships and pertinent study characteristics. This process was designed to collect quantitative data on dietary nutrient intake, associated health outcomes, effect estimates (e.g., relative risks and hazard ratios) across various intake levels, and the corresponding confidence intervals. Additionally, the risk of bias assessment for each meta-analysis, as conducted using the ROBIS tool, was documented.

## 3. Results and Discussion

Our analysis consistently revealed that the assessed nutrient–health outcome and dose–response relationships are not invariably linear, often exhibiting complexities such as nonlinear dose–response curves, threshold effects, and significant modulation by nutrient sources and food matrices. Several components were identified as exhibiting significant dose–response relationships, indicating both protective and potentially adverse effects depending on the intake level and the nutrient source. Upon evaluation, most meta-analyses were determined to have a low risk of bias. However, some studies exhibited a significant risk of bias. Specifically, studies on the dose–response relationship between dietary calcium intake and prostate cancer [14], fibre intake and ovarian cancer [15], haem iron intake and coronary heart disease (CHD) [16] and colorectal cancer [17], zinc intake and colorectal cancer [17], magnesium intake and colorectal cancer [18], saturated fatty acid (SFA) intake and Alzheimer’s disease [19], and selenium intake and type 2 diabetes (T2DM) [20] were identified as having a high risk of bias according to the ROBIS assessment (Figure 2).

In alignment with the principles of risk–benefit assessment outlined by Boué et al. [6], which emphasise a structured approach to component selection, our comprehensive review provides an evidence base for nutrient–health outcome relationships. Furthermore, the practical application of such dose–response data in RBA was exemplified by Ververis et al. [10], who assessed the health impacts of substituting red meat with insects. It should be noted that Ververis et al. [10] conducted a full applied RBA for a specific substitution scenario, calculating net health impacts, whereas our review’s objective is to compile and evaluate existing quantitative dose–response data from meta-analyses to serve as foundational inputs for various RBA contexts. The dose–response relationships and health outcomes identified in our review are consistent with those reported by Ververis et al. [10]. However, while their assessment focused on a selected set of nutrient–outcome associations pertinent to a specific food substitution scenario, our review provides a more comprehensive mapping of the evidence base, encompassing a broader range of health outcomes for each nutrient. For overlapping nutrient–outcome pairs, the directionality of associations (protective or adverse) is generally concordant, and both analyses prioritise evidence from meta-analyses assessed as having a low risk of bias. While specific citations and the level of detail regarding nuances in dose–response relationships differ between studies, reflecting their distinct objectives, both underscore the significance of considering robust meta-analytic evidence for nutrient–health relationships within risk–benefit assessment.

The subsequent subsections provide a comprehensive, nutrient-specific analysis of these findings, presenting evidence for nutrient–health outcome relationships and establishing the foundation for a nuanced discussion of their implications for dietary recommendations and public health interventions. Table 1 provides a detailed summary, listing each nutrient along with the related health outcomes, source of evidence from the respective meta-analyses, and corresponding risk of bias assessments.

### 3.1. Calcium

Meta-analyses of epidemiological studies have consistently demonstrated an inverse association between calcium intake and the risk of several cancers, including breast cancer [21], colorectal cancer [25], and glioma [23]. Furthermore, a higher calcium intake is associated with a reduced risk of stroke [22] and T2DM [24] (Table 2). Notably, one meta-analysis suggested a potential increase in the risk of prostate cancer associated with higher calcium intake [14]. Nevertheless, this correlation seems to be specific to calcium obtained from dairy sources, as calcium from non-dairy sources does not exhibit a similar effect [14]. Moreover, Aune et al. [14] exhibited a high risk of bias owing to the absence of an explicit risk of bias assessment for the included cohort studies. Consequently, the potential correlation between elevated calcium intake and an increased risk of prostate cancer should be approached with considerable caution. This distinction underscores the complexity of nutrient–disease relationships and emphasises the importance of considering food sources when interpreting such associations.

### 3.2. Iron

The association between iron intake and health outcomes appears complex, particularly concerning haem and non-haem iron. While total iron intake has shown a potential inverse association with oesophageal cancer [29], studies have indicated that a higher haem iron intake is associated with an increased risk of breast cancer [28], cardiovascular disease (CVD) [30], CHD [16], colorectal cancer [17], and T2DM [27] (Table 3). Nevertheless, a ROBIS assessment identified a significant risk of bias in the meta-analysis concerning haem iron and CHD [16] and colorectal cancer [17]. This risk is primarily due to the absence of a risk of bias assessment for the included cohort studies; therefore, these findings should be interpreted with caution. Importantly, these elevated risks are primarily associated with haem iron, which is predominantly found in animal-derived products. Associations with non-haem iron were less consistent or absent in numerous studies [16,17,27,28]. These findings suggest that the source of iron, haem versus non-haem, is a crucial factor in determining its health impact, with haem iron potentially presenting a higher risk of certain chronic diseases, particularly cancer, T2DM, and CVD [16,17,27,28,30].

### 3.3. Zinc

Meta-analyses indicated that higher zinc intake may be associated with a reduced risk of Parkinson’s disease [31], oesophageal cancer [29], and colorectal cancer [17] (Table 4). However, one meta-analysis on colorectal cancer [17] identified a U-shaped association, indicating that both very low and very high zinc intakes may be correlated with an elevated risk. The optimal intake level is suggested to be approximately 22 mg/day [17]. This underscores the significance of considering potential nonlinear dose–response relationships for certain nutrients and the concept of optimal intake ranges, rather than merely assuming a linear “more is better” approach. Nonetheless, these findings should be interpreted with caution, as the meta-analysis on zinc and colorectal cancer [17] exhibited a high risk of bias, largely attributable to the lack of risk of bias evaluation for the included studies.

### 3.4. Magnesium

Evidence from meta-analyses indicates a beneficial role for magnesium in reducing the risk of several chronic conditions. Higher magnesium intake has been associated with a decreased risk of depression [32], T2DM [33], stroke [33], CVD [34], CHD [34], hypertension [35], and heart failure [36]. Furthermore, one meta-analysis indicated a potential association with a reduced risk of colorectal cancer [18] (Table 5). Nevertheless, this finding is associated with a considerable risk of bias owing to the lack of a formal risk of bias assessment for the included studies. Therefore, this potential protective association should be interpreted with caution. In addition, similar to the findings concerning fibre, the results of the meta-analysis on magnesium intake and depression [32] should be approached with caution. This is because several cross-sectional studies were included, which cannot establish temporal relationships or causality. Notably, for T2DM risk reduction, the beneficial association appears to be the most pronounced within a specific intake range (from 50 mg/day to 150 mg/day increase), with evidence suggesting a plateauing effect at very high intakes [33]. This suggests that, although beneficial, the dose–response relationship for magnesium may not be linear across the entire range of intakes.

### 3.5. Selenium

One meta-analysis indicated a potential negative impact of increased dietary selenium intake on the risk of T2DM [20] (Table 6). Specifically, selenium consumption exceeding 80 µg/day is associated with an elevated risk of T2DM, with evidence suggesting a threshold effect and a J-shaped relationship with blood selenium levels [20]. However, this meta-analysis did not provide specific risk estimates for each incremental intake of selenium, thereby constraining the precise dose–response quantification. Nevertheless, this finding highlights the importance of considering the potential risks associated with exceeding certain selenium intake levels and suggests a complex dose–response relationship in which a higher intake is not necessarily beneficial. Further research is needed to fully elucidate the optimal selenium intake range and its effects on health outcomes. Nonetheless, this meta-analysis is subject to a significant risk of bias, primarily due to the lack of a formal risk of bias assessment for the included primary studies and limitations in the robustness of the synthesis methods, which raises concerns regarding the reliability of the conclusions. However, current evidence suggests caution regarding high dietary selenium consumption and T2DM risk.

### 3.6. Sodium

A high sodium intake is consistently associated with adverse health outcomes. Meta-analyses indicated a positive association between increased sodium consumption and a higher risk of cardiovascular events [37], hypertension [38], stroke [39], and gastric cancer [40] (Table 7).

### 3.7. Vitamin B12

Evidence regarding the association between vitamin B12 and cancer risk has been mixed. A meta-analysis suggested a potentially increased risk of oesophageal adenocarcinoma with higher vitamin B12 intake [41], while another meta-analysis indicated an inverse association between vitamin B12 and colorectal cancer risk [42] (Table 8). However, a recent scoping review examining the association between vitamin B12 and cancer concluded that there is insufficient evidence to establish a causal relationship between high vitamin B12 intake and cancer [64]. Further investigations are necessary to elucidate the complex relationship between vitamin B12 levels and carcinogenesis.

### 3.8. Vitamin D3

Meta-analyses indicate that a higher vitamin D intake may be associated with a reduced risk of several cancers, including colorectal cancer [25], lung cancer [44], and pancreatic cancer [45]. Moreover, a meta-analysis identified a nonlinear association between vitamin D intake and stroke risk [43] (Table 9). The analysis revealed that the risk of stroke was minimised at an intake of approximately 12 µg/day of vitamin D, corresponding to an approximate 20% reduction in risk, with no further benefits observed at higher intake levels [43]. Nevertheless, this study did not provide specific risk estimates for incremental increases in vitamin D intake, which constitutes a limitation in accurately quantifying dose–response relationships. These findings suggest a potential protective role of adequate vitamin D status in cancer and CVD prevention while also highlighting the possibility of optimal intake levels rather than a continuously increasing benefit.

### 3.9. Fibre

A substantial body of evidence from meta-analyses supports the beneficial role of dietary fibre in various health outcomes. Increased fibre intake has been consistently associated with a reduced risk of colorectal cancer [51], breast cancer [49], liver cancer [46], diverticular disease [52], and T2DM [51] in the general population. Moreover, higher fibre consumption is associated with a lower risk of CVD [51], CHD [51], and stroke [51]. Protective associations have also been identified in ovarian cancer [15], pancreatic cancer [54], oesophageal cancer [55], bladder cancer [50], chronic obstructive pulmonary disease (COPD) [47], Crohn’s disease [53], and depression [48] (Table 10). Nevertheless, one meta-analysis on ovarian cancer [15] identified a significant risk of bias attributable to the lack of a risk of bias assessment for the included cohort studies. Consequently, the finding of a potential protective effect should be interpreted with caution. Furthermore, the results of the meta-analysis on fibre intake and depression [48] should also be approached with caution, as it included several cross-sectional studies, which are unable to establish temporal relationships or causality. Notably, although the collective findings underscore the extensive health benefits associated with fibre-rich diets, these benefits are not consistent across different types of dietary fibre and can vary considerably depending on the specific health outcomes being considered. For instance, in the context of colorectal cancer risk reduction, cereal fibre appears to be the most significantly associated type, with associations for fibre from vegetables, fruits, and legumes being less pronounced or non-significant in some analyses [65]. Conversely, when considering COPD, evidence suggests that fruit fibre may be more relevant for risk reduction, whereas vegetable fibre shows non-significant associations [47]. This underscores the critical role of fibre type in determining its impact on various health outcomes.

### 3.10. Saturated Fatty Acids

The association between SFAs and health outcomes remains a subject of ongoing scientific discourse. While a higher SFA intake has been linked to an increased risk of Alzheimer’s disease [19], liver cancer [57], and endometrial cancer [60], its impact on cardiovascular health appears to be more nuanced and context-dependent. Specifically, the substitution of SFAs with linoleic acid (LA) is associated with a reduction in CHD risk [61] and a decrease in saturated fat intake, particularly when replaced by unsaturated fats, which may further reduce the risk of CVD [59] (Table 11). Although the Cochrane review by Hooper et al. [59] did not provide a single validated threshold for SFA intake and CVD risk, it supported a dose–response relationship: the greater the reduction in SFA intake (resulting in more significant cholesterol lowering), the greater the potential reduction in cardiovascular events [59]. However, certain meta-analyses have indicated that total SFA intake may not be significantly associated with coronary events [66]. Nonetheless, upon examining the types of SFAs and food sources, the consumption of palmitic and stearic acids, as well as SFAs derived from meat sources, may be modestly associated with an increased risk [66]. Conversely, the proposed link between SFA consumption and Alzheimer’s disease should be approached with caution, since the meta-analysis conducted by Ruan et al. [19] exhibited a high risk of bias, primarily attributable to substantial heterogeneity and the lack of assessment of publication bias in the included studies.

Notably, SFA intake appears to have varying effects on health outcomes. Although a meta-analysis indicated a potential reduction in stroke risk associated with increased SFA intake [58], the authors underscored the need for further research to explore how specific types of SFA and various macronutrient substitution models influence stroke risk, as these aspects were not thoroughly examined due to limitations in the available data [58]. Other studies have indicated a potentially protective effect of a lower dietary SFA intake, specifically below 40 g/day, on colorectal cancer [67]. These apparently contradictory findings underscore the significance of considering not only the absolute consumption of SFAs but also the broader nutritional context, particularly the macronutrients with which they are being compared or substituted.

### 3.11. n-3 Fatty Acids

Meta-analyses indicate a protective effect of n-3 fatty acids, particularly alpha-linolenic acid (ALA), against CHD [62] (Table 12). One study suggested an inverse association between ALA intake and prostate cancer risk [68]. However, the evidence was inconclusive and should be interpreted cautiously.

### 3.12. n-6 Fatty Acids

The consumption of n-6 polyunsaturated fatty acids (PUFAs), particularly LA, exhibits cardiovascular benefits, including advantages derived from the substitution of LA for SFAs, as evidenced by a meta-analysis indicating a reduced risk of CHD events [61]. Furthermore, an increased consumption of linoleic acid, an n-6 fatty acid, is associated with a decreased risk of T2DM [63] (Table 13).

### 3.13. Nutrients Not Included

Copper, thiamine, niacin, and monounsaturated fatty acids (MUFAs) were omitted from the final summary table (Table 1) because of the absence of identified dose–response relationships with health outcomes in meta-analyses that satisfied our inclusion criteria. Although research has been conducted on these nutrients, the available meta-analyses either presented associations as high-versus-low intake comparisons or did not demonstrate statistically significant dose–response gradients. As the risk–benefit assessment necessitates dose–response data to model health impacts, these nutrients, which lack such data based on our search strategy, were excluded from the analysis.

### 3.14. Strengths and Limitations

This review has several key methodological strengths. It employed a comprehensive approach, emphasising the synthesis of evidence from recent high-quality meta-analyses. Additionally, the ROBIS tool facilitated a critical evaluation of bias within the included studies, thereby enhancing the robustness of the findings. Importantly, a comprehensive search strategy was implemented across three major databases—PubMed, Scopus, and Web of Science—to maximise the breadth of the identified literature. Despite these strengths, several limitations of systematic reviews and meta-analyses warrant consideration when interpreting these findings. First, the reliance on meta-analyses implies that conclusions are contingent upon the quality of the primary studies analysed, and even with the use of ROBIS, the risk of residual bias cannot be entirely excluded. The identification of several key nutrient–outcome relationships, primarily supported by meta-analyses assessed with a high risk of bias (as noted in the text), necessitates particular caution when employing these specific dose–response data as inputs for quantitative RBA models. This caution is advised pending further confirmation from higher-quality studies. Second, the primary aim of establishing a foundational evidence base for multinutrient RBA required a comprehensive scope encompassing 12 nutrients. This broad scope inherently restricted the depth of analysis feasible for any single nutrient. Consequently, the detailed examination of mechanisms, pharmacokinetics, or effects on specific population subgroups was limited by this RBA-focused approach and the available meta-analytic data. Third, this review was constrained by the absence of detailed population stratification in the source meta-analyses. Nutrient requirements and physiological responses vary significantly based on age, sex, physiological condition (e.g., pregnancy), and genetic factors [69,70]. Dietary recommendations and the interpretation of nutrient adequacy are inherently population-specific [69]. Age was not utilised as an explicit inclusion or exclusion criterion during the selection of meta-analyses for this review, as our aim was to synthesise the quantitative evidence reported in the existing meta-analytic literature suitable for foundational RBA input. However, many of these meta-analyses primarily aggregate data from general adult populations or lack detailed stratification across the lifespan (e.g., specific analyses for children, adolescents, and the elderly) or physiological states necessary for nuanced interpretation. Consequently, the dose–response relationships presented in this study largely reflect findings from broad adult populations and may not be directly generalisable to specific subgroups without further investigation or adjustment. Modern RBA frameworks emphasise the importance of considering population subgroups and variability [6,8], underscoring the caution required when applying the generalised findings compiled in this review to specific subpopulations or individuals in future RBA models. This lack of stratification in the underlying evidence base represents a potential source of uncertainty and confounding bias, limiting the direct applicability of some of the findings. Fourth, although a comprehensive search was conducted across multiple databases, the use of specific search terms may have resulted in the omission of relevant studies indexed with different terminologies or located in more specialised databases. Finally, the predefined nutrient selection, while necessary for managing the scope, indicates that this review is not fully comprehensive in terms of all dietary factors pertinent to the risk–benefit assessment. These limitations underscore the need for further research to augment the evidence base for comprehensive risk–benefit assessments.

## 4. Conclusions

This comprehensive review offers significant insight into the complex dose–response relationships between nutrients and health outcomes, establishing a solid foundation for risk–benefit assessments of dietary modifications. The analysis identified diverse and complex relationships, underscoring the necessity of considering factors beyond the mere presence of nutrients, such as nonlinear dose–response curves (e.g., for zinc), nutrient sources (e.g., haem versus non-haem iron), and potential interactions within dietary patterns. While this review provides a systematically derived foundation, it also highlights data gaps, particularly concerning nutrients such as copper, thiamine, niacin, and MUFAs. The synthesis of these findings highlights the complex nature of nutrient–health interactions and emphasises the necessity for further research to enhance the understanding of dose–response relationships, particularly in areas that are under-researched and for nutrients that lack quantitative data. This compiled evidence serves as an essential quantitative resource for informing future risk–benefit assessments of various food scenarios, including those involving novel alternatives. As research continues to address these identified data gaps, these results will facilitate more precise and evidence-based dietary recommendations and public health strategies, thereby contributing to healthier and more sustainable dietary patterns within evolving global food systems.

## Figures and Tables

**Figure 1 foods-14-01420-f001:**
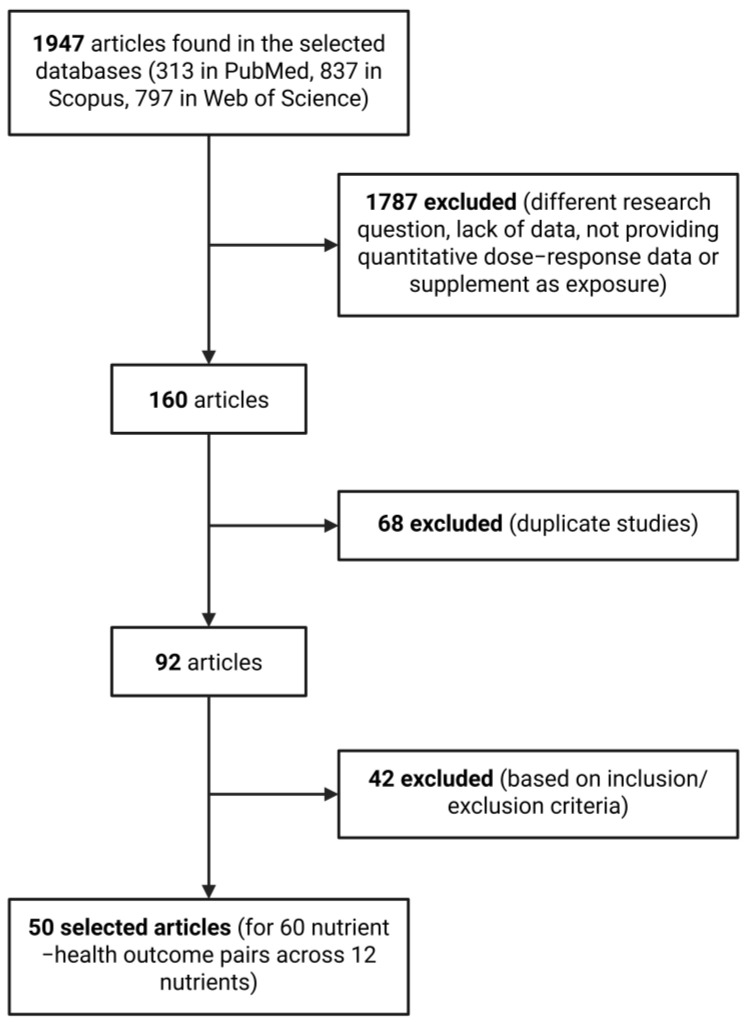
Flowchart of study selection process.

**Figure 2 foods-14-01420-f002:**
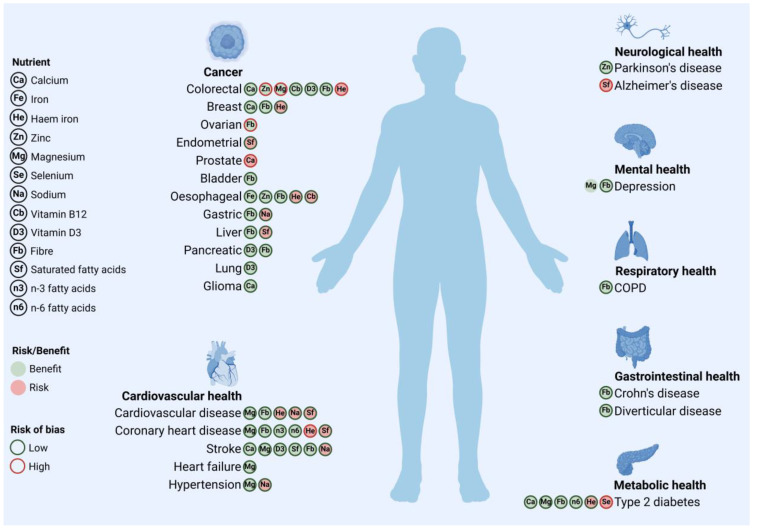
Visual summary of nutrient dose–response associations with various health outcomes. Icon colour indicates potential direction of effect (red for risk, green for benefit), while icon fill denotes assessed risk of bias (low vs. high) from supporting meta-analysis. Specific forms of iron (e.g., haem vs. non-haem) are detailed in text and tables. This figure offers preliminary overview, serving as essential input for risk–benefit assessment. Created in https://BioRender.com.

**Table 1 foods-14-01420-t001:** The health outcomes associated with each of the selected nutrients and the corresponding risk of bias (arrows indicate the direction of association: ↑ = increased risk; ↓ = reduced risk).

Component	Health Outcome (Risk)	Source	Risk of Bias
Calcium	Breast cancer (↓)	[21]	Low
	Stroke (↓)	[22]	Low
	Glioma (↓)	[23]	Low
	T2DM (↓)	[24]	Low
	Colorectal cancer (↓)	[25]	Low
	Hypertension (↓)	[26]	Low
	Prostate cancer (↑)	[14]	High
Iron	T2DM (haem) (↑)	[27]	Low
	Breast cancer (haem) (↑)	[28]	Low
	Oesophageal cancer (total) (↓); (haem) (↑)	[29]	Low
	CVD (haem) (↑)	[30]	Low
	CHD (haem) (↑)	[16]	High
	Colorectal cancer (haem) (↑)	[17]	High
Zinc	Parkinson’s disease (↓)	[31]	Low
	Oesophageal cancer (↓)	[29]	Low
	Colorectal cancer (↓)	[17]	High
Magnesium	Depression (↓)	[32]	n.a. *
	T2DM (↓)	[33]	Low
	Stroke (↓)	[33]	Low
	CVD (↓)	[34]	Low
	CHD (↓)	[34]	Low
	Hypertension (↓)	[35]	Low
	Heart failure (↓)	[36]	Low
	Colorectal cancer (↓)	[18]	High
Selenium	T2DM (↑)	[20]	High
Sodium	CVD (↑)	[37]	Low
	Hypertension (↑)	[38]	Low
	Stroke (↑)	[39]	Low
	Gastric cancer (↑)	[40]	Low
Vitamin B12	Oesophageal cancer (↑)	[41]	Low
	Colorectal cancer (↓)	[42]	Low
Vitamin D3	Colorectal cancer (↓)	[25]	Low
	Stroke (↓)	[43]	Low
	Lung cancer (↓)	[44]	Low
	Pancreatic cancer (↓)	[45]	Low
Fibre	Liver cancer (↓)	[46]	Low
	COPD (↓)	[47]	Low
	Depression (↓)	[48]	Low
	Breast cancer (↓)	[49]	Low
	Bladder cancer (↓)	[50]	Low
	T2DM (↓)	[51]	Low
	Colorectal cancer (↓)	[51]	Low
	Diverticular disease (↓)	[52]	Low
	CVD (↓)	[51]	Low
	CHD (↓)	[51]	Low
	Stroke (↓)	[51]	Low
	Ovarian cancer (↓)	[15]	High
	Crohn’s disease (↓)	[53]	Low
	Pancreatic cancer (↓)	[54]	Low
	Oesophageal cancer (↓)	[55]	Low
	Gastric cancer (↓)	[56]	Low
Saturated fatty acids	Liver cancer (↑)	[57]	Low
	Stroke (↓)	[58]	Low
	CVD (↑)	[59]	Low
	Alzheimer’s disease (↑)	[19]	High
	Endometrial cancer (↑)	[60]	Low
	CHD (↑)	[61]	Low
n-3 fatty acids	CHD (ALA) (↓)	[62]	Low
n-6 fatty acids	T2DM (↓)	[63]	Low
	CHD (↓)	[61]	Low

* Risk of bias not assessed due to lack of full-text access.

**Table 2 foods-14-01420-t002:** Dose–response associations between calcium intake and health outcomes.

Source	Health Outcome	Study	Results (Dose–Response)
[21]	Breast cancer	Meta-analysis (7 cohort studies, *n* = 1,579,904)	Per 350 mg/day increase: BC risk ↓ 6% (RR: 0.94, 0.89–0.99).
[22]	Stroke *	Meta-analysis (18 cohort studies, *n* = 882,181)	Per 200 mg/day increase: stroke risk ↓ 5% (RR: 0.95, 0.92–0.98). Per 300 mg/day increase: stroke risk ↓ 6% (RR: 0.94, 0.90–0.98). Per 500 mg/day increase: stroke risk ↓ 5% (RR: 0.95, 0.90–0.99).
[23]	Glioma	Meta-analysis (4 case–control studies, *n* = 1942)	Per 100 mg/day increase: glioma risk ↓ 7% (OR: 0.93, 0.88–0.98).
[24]	T2DM	Meta-analysis (8 cohort studies, *n* = 255,744)	Per 300 mg/day increase: T2DM risk ↓ 7% (RR: 0.93, 0.89–0.98). Per 600 mg/day increase: T2DM risk ↓ 14% (RR: 0.87, 0.79–0.97). Per 1000 mg/day increase: T2DM risk ↓ 23% (RR: 0.80, 0.67–0.95).
[25]	Colorectal cancer	Meta-analysis (148 cohort studies, 18 RCTs, *n* = 854,195)	Per 400 mg/day increase: CRC risk ↓ 5% (RR: 0.95, 0.94–0.96).
[26]	Hypertension	Meta-analysis (8 cohort studies, *n* = 248,398)	Per 500 mg/day increase: hypertension risk ↓ 7% (RR: 0.93, 0.90–0.97).
[14]	Prostate cancer **	Meta-analysis (9 cohort studies, *n* = 750,275)	Per 400 mg/day increase: prostate cancer risk ↑ 5% (RR: 1.05, 1.02–1.09).

* Only Asian countries. ** No association with non-dairy calcium intake. Arrows indicate the direction of association: ↑ = increased risk; ↓ = reduced risk.

**Table 3 foods-14-01420-t003:** Dose–response associations between iron intake and health outcomes.

Source	Health Outcome	Study	Results (Dose–Response)
[27]	T2DM *	Meta-analysis (11 cohort studies, *n* = 323,788)	Per 1 mg/day (haem) increase: T2DM risk ↑ 16% (RR: 1.16, 1.03–1.30).
[28]	Breast cancer *	Meta-analysis (23 observational studies)	Per 1 mg/day (haem) increase: BC risk ↑ 8% (RR: 1.08, 1.002–1.17).
[29]	Oesophageal cancer	Meta-analysis (20 observational studies, *n* = 1,387,482)	Per 5 mg/day increase: OC risk ↓ 15% (OR: 0.85, 0.79–0.92). Per 1 mg/day (haem) increase: OC risk ↑ 21% (OR: 1.21, 1.02–1.45).
[30]	CVD *	Meta-analysis (13 observational studies, *n* = 252,164)	Per 1 mg/day (haem) increase: CVD risk ↑ 7% (RR: 1.07, 1.01–1.14).
[16]	CHD *	Meta-analysis (6 cohort studies, *n* = 131,553)	Per 1 mg/day (haem) increase: CHD risk ↑ 27% (RR: 1.27, 1.10–1.47).
[17]	Colorectal cancer *	Meta-analysis (8 cohort studies, *n* = 651,272)	Per 1 mg/day (haem) increase: CRC risk ↑ 11% (RR: 1.11, 1.03–1.18).

* Significant association with only haem iron. Arrows indicate the direction of association: ↑ = increased risk; ↓ = reduced risk.

**Table 4 foods-14-01420-t004:** Dose–response associations between zinc intake and health outcomes. Arrows indicate the direction of association: ↓ = reduced risk.

Source	Health Outcome	Study	Results (Dose–Response)
[31]	Parkinson’s disease	Meta-analysis (6 case–control studies, 7 cohort studies, *n* = 467,958)	Per 1 mg/day increase: PD risk ↓ 35% (OR: 0.65, 0.49–0.86).
[29]	Oesophageal cancer	Meta-analysis (20 observational studies, *n* = 1,387,482)	Per 5 mg/day increase: OC risk ↓ 15% (OR: 0.85, 0.77–0.93).
[17]	Colorectal cancer	Meta-analysis (6 cohort studies, *n* = 350,507)	Per 5 mg/day increase: CRC risk ↓ 14% (RR: 0.86, 0.78–0.96).

**Table 5 foods-14-01420-t005:** Dose–response associations between magnesium intake and health outcomes. Arrows indicate the direction of association: ↓ = reduced risk.

Source	Health Outcome	Study	Results (Dose–Response)
[32]	Depression	Meta-analysis (10 cross-sectional studies, 3 cohort studies, *n* = 63,214)	Per 100 mg/day increase: depression risk ↓ 7% (RR: 0.93, 0.90–0.96).
[33]	T2DM	Meta-analysis (35 cohort studies, *n* = 1,219,636)	Per <50 mg/day increase: T2D risk ↓ 10% (RR: 0.90, 0.88–0.93). Per ≥50 to <100 mg/day increase: T2D risk ↓ 16% (RR: 0.84, 0.82–0.87). Per ≥100 to <150 mg/day increase: T2D risk ↓ 22% (RR: 0.78, 0.74–0.83). Per ≥150 mg/day increase: T2D risk ↓ 21% (RR: 0.79, 0.74–0.84).
[33]	Stroke	Meta-analysis (18 cohort studies, *n* = 692,998)	Per ≥150 mg/day increase: total stroke risk ↓ 15% (RR: 0.85, 0.79–0.91).
[34]	CVD	Meta-analysis (18 cohort studies, *n* = 544,581)	Per 100 mg/day increase: CVD risk ↓ 10% (RR: 0.90, 0.83–0.96).
[34]	CHD	Meta-analysis (18 cohort studies, *n* = 544,581)	Per 100 mg/day increase: CHD risk ↓ 8% (RR: 0.92, 0.82–0.98).
[35]	Hypertension	Meta-analysis (10 cohort studies, *n* = 180,566)	Per 100 mg/day increase: hypertension risk ↓ 5% (RR: 0.95, 0.90–1.00).
[36]	Heart Failure	Meta-analysis (40 cohort studies, *n* > 1,000,000)	Per 100 mg/day increase: heart failure risk ↓ 22% (RR: 0.78, 0.69–0.89).
[18]	Colorectal cancer	Meta-analysis (8 cohort studies, *n* = 338,979)	Per 50 mg/day increase: CRC risk ↓ 5% (RR: 0.95, 0.89–1.00); colon cancer risk ↓ 7% (RR: 0.93, 0.88–0.99).

**Table 6 foods-14-01420-t006:** Dose–response associations between selenium intake and health outcomes. Arrows indicate the direction of association: ↑ = increased risk.

Source	Health Outcome	Study	Results
[20]	T2DM	Meta-analysis (7 case–control studies, 9 cohort studies, 18 cross-sectional studies)	Compared to 55 μg/day selenium intake: At 80 μg/day, T2DM risk ↑ 23% (RR: 1.23, 1.14–1.33). At 120 μg/day, T2DM risk ↑ 55% (RR: 1.55, 1.27–1.90).

**Table 7 foods-14-01420-t007:** Dose–response associations between sodium intake and health outcomes. Arrows indicate the direction of association: ↑ = increased risk.

Source	Health Outcome	Study	Results (Dose–Response)
[37]	CVD	Meta-analysis (9 observational studies, *n* = 645,006)	Per 1 g/day increase: CVD risk ↑ up to 4% (RR: 1.04; 1.01, 1.07).
[38]	Hypertension	Meta-analysis (11 cohort studies)	Per 4 g/day increase (vs. 2 g/day): hypertension risk ↑ 4% (RR: 1.04, 0.96–1.13). Per 6 g/day increase (vs. 2 g/day): hypertension risk ↑ 21% (RR: 1.21, 1.06–1.37).
[39]	Stroke	Meta-analysis (14 cohort studies, 1 case–cohort study, 1 case–control study, *n* = 261,732)	Per 1 g/day increase: stroke risk ↑ 6% (RR: 1.06, 1.02–1.10).
[40]	Gastric cancer	Meta-analysis (76 cohort studies, *n* = 6,316,385)	Per 5 g/day increase: gastric cancer risk ↑ 12% (RR: 1.12, 95% CI: 1.02 to 1.23).

**Table 8 foods-14-01420-t008:** Dose–response associations between vitamin B12 intake and health outcomes. Arrows indicate the direction of association: ↑ = increased risk; ↓ = reduced risk.

Source	Health Outcome	Study	Results
[41]	Oesophageal cancer	Meta-analysis (26 observational studies, *n* = 510,954)	Per 1 µg/day increase: OC risk ↑ 2% (OR: 1.02 (1.00–1.03).
[42]	Colorectal cancer	Meta-analysis (17 observational studies, *n* = 10,601)	Per 4.5 µg/day increase: CRC risk ↓ 8.6% (RR: 0.914, 0.856–0.977).

**Table 9 foods-14-01420-t009:** Dose–response associations between vitamin D3 intake and health outcomes. Arrows indicate the direction of association: ↓ = reduced risk.

Source	Health Outcome	Study	Results
[25]	Colorectal cancer	Meta-analysis (148 observational studies, 18 RCTs)	Per 200 IU/day increase: CRC risk ↓ 5% (RR: 0.95, 0.92–0.98).
[43]	Stroke	Meta-analysis (20 cohort studies, *n* = 217,235)	High vs. low vitamin D intake: stroke risk ↓ 25% (RR: 0.75, 0.57–0.98). Optimal intake: 12 µg/day for max 20% reduction.
[44]	Lung cancer	Meta-analysis (5 case–control studies, 11 cohort studies, *n* = 280,127)	Per 100 IU/day increase: lung cancer risk ↓ 2.4% (RR: 0.976, 0.957–0.995).
[45]	Pancreatic cancer	Meta-analysis (14 case–control studies, 9 cohort studies, 2 RCTs, *n* = 1,213,821)	Per 10μg/day intake: pancreatic cancer risk ↓ 25% (RR: 0.75, 0.60–0.93).

**Table 10 foods-14-01420-t010:** Dose–response associations between fibre intake and health outcomes. Arrows indicate the direction of association: ↓ = reduced risk.

Source	Health Outcome	Study	Results (Dose–Response)
[46]	Liver cancer	Meta-analysis (7 cohort studies, *n* = 137,481)	Per 10 g/day increase: liver cancer risk ↓ 17% (HR: 0.83, 0.76–0.91).
[47]	COPD	Meta-analysis (5 cohort studies, *n* = 213,912)	Per 10 g/day increase (total dietary fibre): COPD risk ↓ 26% (RR: 0.74, 0.67–0.82). Per 10 g/day increase (cereal fibre): COPD risk ↓ 21% (RR: 0.79, 0.74–0.84). Per 10 g/day increase (fruit fibre): COPD risk ↓ 37% (RR: 0.63, 0.53–0.75).
[48]	Depression	Meta-analysis (12 cross-sectional studies, 5 cohort studies, 1 case–control study)	Per 5 g/day increase: depression risk ↓ 5% (OR: 0.95, 0.94–0.97).
[49]	Breast cancer	Meta-analysis (51 cohort studies, *n* = 2,725,657)	Per 10 g/day increase: BC risk ↓ 3%; premenopausal BC risk ↓ 14%.
[50]	Bladder cancer	Meta-analysis (13 cohort studies, *n* = 574,726)	Per 5 g/day increase: bladder cancer risk ↓ 4% (HR: 0.96, 0.94–0.98).
[52]	Diverticular disease	Meta-analysis (5 cohort studies, *n* = 865,829)	Per 10 g/day increase: diverticular disease risk ↓ 26% (RR: 0.74, 0.71–0.78).
[51]	T2DM	Meta-analysis (185 cohort studies, 58 RCTs, *n* = 865,829)	Per 8 g/day increase: T2DM risk ↓ 15% (HR: 0.85, 0.82–0.89).
[51]	CVD	Meta-analysis (185 cohort studies, 58 RCTs, *n* = 865,829)	Per 8 g/day increase: CVD risk ↓ 22% (HR: 0.78, 0.68–0.90).
[51]	CHD	Meta-analysis (185 cohort studies, 58 RCTs, *n* = 865,829)	Per 8 g/day increase: CHD risk ↓ 19% (HR: 0.81, 0.73–0.90).
[51]	Stroke	Meta-analysis (185 cohort studies, 58 RCTs, *n* = 865,829)	Per 8 g/day increase: stroke risk ↓ 10% (HR: 0.90, 0.85–0.95).
[51]	Colorectal cancer	Meta-analysis (185 cohort studies, 58 RCTs, *n* = 865,829)	Per 8 g/day increase: CRC risk ↓ 8% (HR: 0.92, 0.89–0.95).
[15]	Ovarian cancer	Meta-analysis (14 case–control studies, 5 cohort studies, *n* = 567,742)	Per 5 g/day increase: ovarian cancer risk ↓ 3% (RR: 0.97, 0.95–0.99).
[54]	Pancreatic cancer	Meta-analysis (13 case–control studies, 1 cohort study)	Per 10 g/day increase: pancreatic cancer risk ↓ 12% (OR: 0.88, 0.84–0.92).
[55]	Oesophageal cancer	Meta-analysis (15 case–control studies, *n* = 16,885)	Per 10 g/day increase: oesophageal cancer risk ↓ 31% (OR: 0.69, 0.61–0.79.
[53]	Crohn’s disease	Meta-analysis (6 case–control studies, 2 cohort studies, *n* = 478,604)	Per 10 g/d increase: CD risk ↓ 13% (RR: 0.87, 0.76–0.98).
[56]	Gastric cancer	Meta-analysis (19 case–control studies, 2 cohort studies, *n* = 580,064)	Per 10 g/day increase: gastric cancer risk ↓ 44%.

**Table 11 foods-14-01420-t011:** Dose–response associations between saturated fatty acid intake and health outcomes.

Source	Health Outcome	Study	Results (Dose–Response)
[57]	Liver cancer	Meta-analysis (14 cohort studies)	Per 1% energy increase: liver cancer risk ↑ 4% (RR: 1.04, 1.01–1.07).
[58]	Stroke	Meta-analysis (12 cohort studies, *n* = 462,268)	Per 10 g/day increase: stroke risk ↓ 6% (RR: 0.94, 0.89–0.98).
[59]	CVD *	Meta-analysis (15 RCTs, *n* = 56,675)	Reduced SFA intake vs. higher SFA intake: cardiovascular event risk potentially ↓ 17% (RR 0.79, 0.66–0.93), with greater reductions possibly yielding larger benefits, especially when considering less than 10% of energy. PUFA replacement vs. SFA: cardiovascular event risk ↓ 21%.
[19]	Alzheimer’s disease	Meta-analysis (4 cohort studies, *n* = 8630)	Per 4 g/day increase: AD risk ↑ 15% (RR: 1.15, 1.01–1.31).
[60]	Endometrial cancer	Meta-analysis (14 case–control studies, 7 cohort studies, *n* = 524,583)	Per 10 g/1000 kcal increase: endometrial cancer risk ↑ 17%.
[61]	CHD	Meta-analysis (13 cohort studies, *n* = 310,602)	Per 5% energy intake increase when LA substitutes SFAs: CHD risk ↓ 9% (RR: 0.91, 0.87–0.96).

* Although no dose–response relationship was assessed, greater reductions in SFA intake were associated with larger reductions in the incidence of cardiovascular events. Arrows indicate the direction of association: ↑ = increased risk; ↓ = reduced risk.

**Table 12 foods-14-01420-t012:** Dose–response associations between n-3 fatty acid intake and health outcomes. Arrows indicate the direction of association: ↓ = reduced risk.

Citation	Health Outcome	Study	Results (Dose–Response)
[62]	CHD	Meta-analysis (13 cohort studies, *n* = 345,202)	Per 1 g/day (ALA) increase: fatal CHD risk ↓ 12% (RR: 0.88, 0.81–0.96). Only ALA intake <1.4 g/d had composite CHD risk ↓.

**Table 13 foods-14-01420-t013:** Dose–response associations between n-6 fatty acid intake and health outcomes. Arrows indicate the direction of association: ↓ = reduced risk.

Source	Health Outcome	Study	Results (Dose–Response)
[63]	T2DM	Meta-analysis (31 cohort studies, *n* = 297,685)	Per 5% energy from LA increase: T2DM risk ↓ 10% (RR: 0.90, 0.84–0.98).
[61]	CHD	Meta-analysis (13 cohort studies, *n* = 310,602)	Per 5% energy from LA increase: CHD risk ↓ 10% (RR: 0.90, 0.85–0.94). Per 5% energy intake increase when LA substitutes SFAs: CHD risk ↓ 9% (RR: 0.91, 0.87–0.96).

## Data Availability

The original contributions presented in the study are included in the article/Appendix A. Further inquiries can be directed to the corresponding author.

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
