# Peer review of "Mapping the Quantitative Dose–Response Relationships Between Nutrients and Health Outcomes to Inform Food Risk–Benefit Assessment"

_foods, 2025, doi:10.3390/foods14081420_

Round 1

Reviewer 1 Report

Comments and Suggestions for Authors

This paper attempts to provide a comprehensive review of dose-response relationships between various nutrients and health outcomes, but requires substantial revision. The primary issue is that the scope is excessively broad, resulting in content that is extensive but lacks sufficient depth. The authors cover 12 nutrients and 60 nutrient-health outcome pairs in a single paper, which demonstrates extensive literature collection capabilities but fails to provide adequately in-depth analysis for any specific nutrient. I recommend that the authors narrow the scope to a single category of nutrients to allow for a more thorough and comprehensive analysis.

Line 14-31: The abstract reveals an overly broad scope, covering 12 nutrients and 60 nutrient-health outcome pairs. This expansive approach restricts the depth of analysis possible for any individual nutrient. I suggest narrowing the research focus to a specific nutrient category to enable more thorough analysis.

Line 35-77: The introduction fails to establish a clear conceptual framework and does not adequately explain how various nutrient dose-response relationships can be meaningfully integrated into risk-benefit assessment. A more focused theoretical foundation would significantly enhance the quality of the paper.

Line 78-90: The methodology section lacks detailed description of how methodological heterogeneity between included meta-analyses was addressed. The specific procedures for handling differences between studies examining the same nutrient-outcome relationships should be elaborated.

Line 127-138: The results indicate selection of 12 nutrients with 60 nutrient-health outcome pairs, confirming that the research scope is too broad to conduct in-depth analysis of individual nutrients or mechanisms.

Line 141-154: While ROBIS was appropriately used to assess bias risk, this section fails to adequately explain how bias differences affect result interpretation. A more systematic approach is needed to weight evidence based on methodological quality.

Line 156-158: Figure 2 attempts to summarize all nutrient-health relationships in a single visual, reflecting the overly ambitious scope that prevents detailed mechanistic analysis of any specific relationship.

Line 159-174: The discussion comparing results with Ververis et al. lacks critical analysis of how different methodological approaches might affect the comparability of results. A more thorough examination of methodological differences would strengthen this comparison.

Line 186-200: The calcium section lists associations with health outcomes but provides insufficient exploration of calcium absorption mechanisms, vitamin D interactions, or population-specific variations—all of which are necessary depth elements for risk-benefit assessment.

Line 201-218: The iron section distinguishes between heme and non-heme iron but fails to adequately explain the biological mechanisms underlying different absorption and health effects, limiting the practical application value of these findings.

Line 236-252: The magnesium section lists associations with multiple health outcomes but fails to integrate these findings to provide a complete assessment of magnesium's overall risk-benefit profile. This pattern of listing associations without synthesis is present throughout all nutrient-specific sections.

Line 253-269: The selenium section notes a J-shaped relationship with T2DM but provides insufficient exploration of potential biological mechanisms explaining this non-linear relationship, exemplifying the paper's general lack of in-depth mechanistic analysis.

Line 276-284: The vitamin B12 section acknowledges contradictory evidence but fails to analyze the methodological sources of these contradictions, indicating inadequate handling of heterogeneous evidence.

Line 299-322: The dietary fiber section covers 14 health outcomes but fails to adequately distinguish between physiological effects of different fiber types (soluble, insoluble, fermentable), illustrating how the broad scope impedes necessary detailed analysis.

Line 323-354: The saturated fatty acids section mentions contradictory associations (reduced stroke risk but increased cardiovascular disease risk) without providing a satisfactory integrated explanation for these apparent contradictions.

Line 370-378: The omission of copper, thiamine, niacin, and monounsaturated fatty acids is briefly mentioned, but lacks adequate discussion of current evidence status or research gaps for these nutrients, further reflecting the limitations of the overly broad approach.

Line 397-413: The conclusion is too general and fails to provide specific guidance on how to translate the review findings into concrete risk-benefit assessment models. A more focused study would enable more actionable conclusions.

Author Response

Comments 1: This paper attempts to provide a comprehensive review of dose-response relationships between various nutrients and health outcomes but requires substantial revision. The primary issue is that the scope is excessively broad, resulting in content that is extensive but lacks sufficient depth. The authors cover 12 nutrients and 60 nutrient-health outcome pairs in a single paper, which demonstrates extensive literature collection capabilities but fails to provide adequately in-depth analysis for any specific nutrient. I recommend that the authors narrow the scope to a single category of nutrients to allow for a more thorough and comprehensive analysis.

Response: We acknowledge the reviewer's accurate observation regarding the comprehensive nature of our review, which encompasses 12 nutrients and 60 outcome pairs. We wish to respectfully clarify that this breadth was a deliberate methodological choice, directly driven by the manuscript's primary and explicitly stated objective: to provide a consolidated overview of available quantitative dose-response relationships to serve as an evidence base for the Risk-Benefit Assessment (RBA) of dietary modifications, particularly those involving novel foods.

As outlined in the Introduction (Lines 49-75), RBA methodologies often require the simultaneous consideration of multiple nutrients whose intake levels may change significantly within a specific dietary scenario (e.g. substituting conventional meat with alternative proteins such as insects or shifting towards plant-based diets), having, as much as possible, a holistic overview of the potential health outcomes. Effective RBA modelling relies on accessible dose-response data across this range of potentially impacted nutrients to parameterise the assessment and estimate the net health effect. Therefore, our goal was to systematically gather and present these quantitative associations (effect estimates per unit of intake) from recent high-quality meta-analyses to create a foundational dataset for RBA applications.

This objective differs significantly from that of a traditional nutrient review, which focuses on deep mechanistic analysis, absorption pathways, physiological roles, or detailed synthesis of a complete risk-benefit profile for a single nutrient or class. Such reviews are undoubtedly valuable but require a different and much narrower scope than the present review. Our focus was intentionally on the breadth of quantitative dose-response data needed for RBA, rather than the depth of mechanistic understanding for each individual nutrient. We aimed to summarise the quantitative relationships established in recent literature suitable for RBA input, rather than exhaustively exploring why these relationships exist or delving into factors such as absorption mechanisms or population variations, which often lack readily available quantitative dose-response data from meta-analyses.

While we maintain that the scope aligns with the stated RBA-focused objective, we understand the reviewer's point that this specific purpose and its distinction from other review types could be made more explicit. To address this issue, we have implemented the following revisions:

  • Abstract and introduction enhanced: We have revised the Abstract (lines 14-37, specifically lines 17-20,33-34) and Introduction (lines 43-102, specifically lines 78-85) to more clearly define the review's specific purpose – compiling quantitative dose-response data as foundational input for RBA – and to explicitly state that deep mechanistic analysis, exploration of absorption factors, or detailed population variations for each nutrient are outside the intended scope of this overview.
  • Limitations section strengthened: We have added a statement to the Limitations section (lines 428-433) explicitly acknowledging that the breadth required to serve as a foundational resource for multi-nutrient RBA scenarios inherently limits the analytical depth and mechanistic exploration achievable for any single nutrient within this review.
  • Figure 2 caption: The caption for Figure 2 (line 176) has been slightly revised to reiterate that it provides a high-level summary of the identified associations relevant for initial RBA screening, rather than a detailed mechanistic map.

We hope that this clarification and the accompanying revisions adequately explain the rationale behind the chosen scope in the context of the manuscript's specific aim to support the development and application of RBA. We believe that this overview format, despite its necessary breadth, fills a specific gap by providing a resource for researchers and practitioners in the RBA field.

Comments 2: Line 14-31: The abstract reveals an overly broad scope, covering 12 nutrients and 60 nutrient-health outcome pairs. This expansive approach restricts the depth of analysis possible for any individual nutrient. I suggest narrowing the research focus to a specific nutrient category to enable more thorough analysis.

Response: We concur with the reviewer that the abstract accurately encapsulates the comprehensive scope of this review. As elaborated extensively in our response to the reviewer's primary comment concerning the overall scope (please refer to the first section of our response above), this breadth was deliberately selected to fulfil the specific objective of compiling foundational quantitative dose-response data necessary for the parameterisation of multi-component Risk-Benefit Assessment (RBA).

While we acknowledge that this approach inherently restricts the depth of analysis for any single nutrient—a point now explicitly stated in the revised Limitations section (Lines 428-431)—narrowing the scope to a single nutrient category would hinder the review from achieving its primary purpose of providing a broad evidence base essential for RBA scenarios, which frequently involve simultaneous changes across multiple nutrient categories (e.g. in dietary shifts or novel food introductions).

To ensure that the abstract more clearly communicates this specific RBA-focused purpose and manages expectations regarding analytical depth, we have revised the abstract (specifically lines 17-18 and 33-34). The revised text now explicitly states that the review's aim is to compile quantitative data as foundational input for RBA and clarifies that deep mechanistic analysis is beyond its current scope, thereby contextualising the breadth of the presented data. We believe that these revisions will help appropriately frame the abstract's content within the review's specific objectives.

Comments 3: Line 35-77: The introduction fails to establish a clear conceptual framework and does not adequately explain how various nutrient dose-response relationships can be meaningfully integrated into risk-benefit assessment. A more focused theoretical foundation would significantly enhance the quality of the paper.

Response: We express our gratitude to the reviewer for their insightful comments and appreciate the opportunity to elucidate the conceptual connection between nutrient dose-response relationships and the RBA framework, as presented in our introduction.

We respectfully assert that quantitative dose-response relationships do not merely integrate into RBA; rather, they constitute the fundamental basis upon which the quantitative assessment of both risks and benefits is constructed within the established RBA methodologies.

Our Introduction reflects this established understanding. Specifically, in lines 62-65, we emphasize the RBA methodology's reliance on the "rigorous establishment of nutrient dose-response relationships, specifically the quantitative association between dietary intake and health outcomes," noting that this is "crucial for predictive modelling, accurate assessment of dietary modification impacts..." This statement is grounded in the core principles of the RBA frameworks developed and utilised over the past decade.

Standard RBA frameworks, such as those described by EFSA (2010, updated in EFSA Scientific Committee, 2024), Boué et al. (2015, 2022), and Assunção et al. (2019), delineate a stepwise process. A critical step in these frameworks is the characterisation of health effects (both adverse and beneficial). This characterisation inherently relies on understanding and quantifying the dose-response relationship, that is, how the likelihood or magnitude of an effect changes with the level of exposure or intake (EFSA Scientific Committee, 2024, Section 4.4; Boué et al., 2022, Step 3.1). Without establishing these quantitative dose-response links, estimating quantitatively the actual impact (positive or negative) of different intake scenarios is impossible.

The integration step in RBA, where risks and benefits are compared or combined (often using metrics such as DALYs or QALYs, or through qualitative/semi-quantitative comparisons), can only occur after the individual risks and benefits have been quantitatively characterised based on their respective dose-response relationships (Assunção et al., 2019, Steps 4/6 and 5/6; EFSA Scientific Committee, 2024, Section 4.7). Dose-response data provide the essential input needed to estimate, for example, the number of disease cases prevented (benefit) or caused (risk) under different dietary scenarios, which are then integrated using the chosen RBA metric. The "bottom-up" approach described by Assunção et al. (2019), commonly used for hazards, explicitly relies on "dose-response models”, while the "top-down" approach, often used for nutrients, implicitly uses population-level dose-response association data to estimate attributable cases.

Therefore, our manuscript presents these dose-response relationships not as elements to be integrated into an existing RBA structure but as the foundational quantitative evidence required by the RBA structure itself to function effectively for estimating net health impacts.

However, we acknowledge the reviewer's point that this foundational role could be articulated more clearly. To enhance this, we propose adding a sentence to the revised Introduction (lines 66-68) to explicitly state that the characterisation of health effects via dose-response analysis is a prerequisite step within standard RBA frameworks (citing, for example, EFSA Scientific Committee, 2024; Boué et al., 2022) that enables the subsequent comparison and integration of risks and benefits.

We believe that this clarification reinforces our review, and by compiling these essential dose-response data, directly supports the core quantitative requirements of the established RBA methodology.

References:

Assunção, R.; Alvito, P.; Brazão, R.; Carmona, P.; Fernandes, P.; Jakobsen, L.S.; Lopes, C.; Martins, C.; Membré, J.-M.; Monteiro, S.; et al. Building Capacity in Risk-Benefit Assessment of Foods: Lessons Learned from the RB4EU Project. Trends in Food Science & Technology 2019, 91, 541–548, doi:10.1016/j.tifs.2019.07.028.

Boué, G.; Guillou, S.; Antignac, J.-P.; Bizec, B.L.; Membré, J.-M. Public Health Risk-Benefit Assessment Associated with Food Consumption–A Review. European Journal of Nutrition & Food Safety 2015, 32–58, doi:10.9734/EJNFS/2015/12285.

Boué, G.; Ververis, E.; Niforou, A.; Federighi, M.; Pires, S.M.; Poulsen, M.; Thomsen, S.T.; Naska, A. Risk-Benefit Assessment of Foods: Development of a Methodological Framework for the Harmonized Selection of Nutritional, Microbiological, and Toxicological Components. Front Nutr 2022, 9, 951369, doi:10.3389/fnut.2022.951369.

EFSA Scientific Committee; More, S.J.; Benford, D.; Hougaard Bennekou, S.; Bampidis, V.; Bragard, C.; Halldorsson, T.I.; Hernández-Jerez, A.F.; Koutsoumanis, K.; Lambré, C.; et al. Guidance on Risk-Benefit Assessment of Foods. EFSA J 2024, 22, e8875, doi:10.2903/j.efsa.2024.8875.

Comments 4: Line 78-90: The methodology section lacks detailed description of how methodological heterogeneity between included meta-analyses was addressed. Specific procedures for handling differences between studies examining the same nutrient-outcome relationships should be elaborated.

Response: We thank the reviewer for highlighting the need for further clarification on how we handled situations in which multiple meta-analyses addressed the same nutrient-outcome pair, potentially exhibiting methodological heterogeneity. We acknowledge that the specific procedure for selecting among these was not explicitly detailed in the methodology section of the original manuscript.

Our approach was guided by the inclusion criteria outlined in Section 2.2, which prioritised evidence from the most methodologically sound and current sources. When multiple eligible meta-analyses were identified for a specific nutrient-outcome pair through our search, the following hierarchical procedure was applied to select the primary source for inclusion in our review:

  • Risk of Bias Assessment: Strong preference was given to meta-analyses with a low risk of bias, as assessed using the ROBIS tool [13]. Meta-analyses with a high risk of bias were generally excluded unless no low-bias alternative was available for a critical nutrient-outcome pair relevant to the review's scope.
  • Date of publication: Among studies with a comparable (low) risk of bias, preference was given to the most recent meta-analysis to ensure the inclusion of the latest available primary studies.
  • Data suitability: The presence and clarity of quantitative dose-response data (e.g. effect estimates per defined intake increment with confidence intervals) suitable for potential use in RBA modelling was also a key consideration.

Therefore, it is important to clarify the scope of our review. Our aim was to review and synthesise the conclusions of existing high-quality meta-analyses rather than reanalyse the primary studies or conduct a new meta-analysis. Therefore, a formal quantitative assessment of methodological heterogeneity between different eligible meta-analyses (e.g. through meta-regression or detailed comparison of the included study populations/methods) was not performed and was considered beyond the scope of this study. Our procedure focused on identifying and selecting the single most appropriate published meta-analysis for each nutrient-outcome pair based on the predefined quality (ROBIS), recency, and data suitability criteria.

To address the reviewer's comment, we added clarifying sentences to Section 2.2 (Inclusion Criteria – lines 122-126) in the revised manuscript to explicitly describe the selection hierarchy employed when multiple eligible meta-analyses were retrieved for the same nutrient-outcome relationship.

Comments 5: Line 127-138: The results indicate selection of 12 nutrients with 60 nutrient-health outcome pairs, confirming that the research scope is too broad to conduct in-depth analysis of individual nutrients or mechanisms.

Response: We agree with the reviewer that the number of nutrient-outcome pairs presented in the results (lines 148-154) reflects the comprehensive scope chosen for this review. As previously elaborated in our initial response regarding the overall scope, this breadth was a deliberate choice driven by the specific aim of compiling the foundational quantitative data necessary for a multi-component Risk-Benefit Assessment (RBA). We again refer the reviewer to the detailed justification provided earlier. Consequently, while the results section accurately reflects the output of our systematic search aimed at informing RBA, we acknowledge, as stated in the revised Limitations section (lines 428-431), that in-depth analysis of mechanisms for each individual nutrient was not the primary objective and is inherently limited by this RBA-focused scope.

Comments 6: Line 141-154: While ROBIS was appropriately used to assess bias risk, this section fails to adequately explain how bias differences affect result interpretation. A more systematic approach is needed to weight evidence based on methodological quality.

Response: We thank the reviewer for this comment and agree that the implications of the ROBIS risk of bias assessments for interpreting the review's findings could be more explicitly articulated in some cases.

Our primary use of the ROBIS tool was twofold: 1) as a key criterion for selecting the most robust meta-analyses for inclusion (prioritising low-bias studies, as described in Section 2.3), and 2) as a critical qualifier for the evidence presented. Although we did not apply a formal quantitative evidence weighting system (e.g. GRADE), the ROBIS assessment directly informed our confidence in specific findings.

Upon review, we noted that the original manuscript already included explicit statements advising caution when interpreting findings derived from meta-analyses with a high risk of bias for several key nutrient-outcome pairs, including haem iron/CHD [16], haem iron/colorectal cancer [17], zinc/colorectal cancer [17], selenium/T2DM [20], and SFA/Alzheimer’s disease [19] (see Lines 236-240, 257-260, 291-294, 369-372 respectively).

However, we agree that this could be applied even more consistently and explicitly across all instances where high-bias studies were discussed (namely, for calcium/prostate cancer [14], fibre/ovarian cancer [15], and magnesium/colorectal cancer [18], where the high bias was noted but the explicit call for caution was less direct).

To address this and ensure maximum clarity and a systematic approach, we made the following revisions:

  • We reviewed the text and ensured that all nutrient-outcome sections referencing a high-risk-of-bias meta-analysis now include a clear and explicit statement advising the reader to interpret those specific findings with caution owing to the identified methodological limitations (lines 220-222; 339-240; 269-270).
  • We have added a statement to the Limitations section (Section 3.14) to generally discuss how the presence of both low- and high-bias studies influences the overall confidence in the review's conclusions and their use in RBA.

We believe that these revisions fully address the reviewer's concerns by explicitly and systematically linking the ROBIS assessment to the interpretation of findings throughout the manuscript.

Comments 7: Line 156-158: Figure 2 attempts to summarize all nutrient-health relationships in a single visual, reflecting the overly ambitious scope that prevents detailed mechanistic analysis of any specific relationship.

Response: We agree with the reviewer that Figure 2 provides a high-level visual summary of the comprehensive scope of our review. As discussed in our primary response regarding the overall scope, the review's objective was to compile a broad overview of quantitative dose-response relationships relevant for Risk-Benefit Assessment (RBA), rather than to conduct a detailed mechanistic analysis for each nutrient-outcome pair.

Consequently, Figure 2 serves as a visual index or map of the key associations identified across the included nutrients and health outcomes, indicating the direction of the association (potential risk/benefit) and the assessed risk of bias from the underlying meta-analysis. It primarily functions as a screening tool to quickly identify areas of interest or concern within the RBA context, complementing the detailed information provided in the text and tables. It was not designed to convey mechanistic details, which, as the reviewer rightly points out, would be unfeasible given the figure's summary nature and the review's RBA-focused scope of the review.

To ensure the figure's purpose is clearly understood, we have, as mentioned previously, revised the caption for Figure 2 (line 176) to explicitly state that it offers a high-level summary relevant for initial RBA screening and is not intended as a detailed mechanistic map. We believe that this clarification aligns the interpretation of the figure with the review's stated objectives and limitations.

Comments 8: Line 159-174: The discussion comparing results with Ververis et al. lacks critical analysis of how different methodological approaches might affect the comparability of results. A more thorough examination of methodological differences would strengthen this comparison.

Response: We thank the reviewer for this valuable suggestion and agree that explicitly outlining the methodological distinctions between our review and the study by Ververis et al. would strengthen the comparison presented in lines 190-201.

As discussed in the overall scope of our work, the primary objective of this review was to identify, compile, and evaluate existing quantitative dose-response relationships between various nutrients and health outcomes derived from recent high-quality meta-analyses. The aim was to provide a broad, foundational evidence base that can serve as reliable input data for RBAs across diverse dietary modification scenarios, including those involving novel foods.  

In contrast, Ververis et al. conducted a full, applied RBA focused on a single, specific substitution scenario. Their methodology involves integrating specific food composition data, national dietary consumption data, exposure assessments, and calculating the net health impact using disability-adjusted DALYs for a particular scenario.  

Given these distinct objectives and methodological approaches (i.e. compiling foundational dose-response data vs. conducting a full RBA for a specific case), the comparison in our manuscript focuses primarily on the concordance of the underlying nutrient-health outcome associations identified and the consistency in prioritising evidence from low-bias meta-analyses where applicable. Our review provides foundational evidence that informs and is utilised in applied assessments, such as the one performed by Ververis et al.

To address the reviewer's comment and enhance clarity, we have revised the manuscript (lines 186-201) as follows:

  • We added text specifying that our review compiles foundational dose-response data (RBA inputs) from meta-analyses, whereas Ververis et al. performed a full applied RBA calculating net health impacts for a specific substitution scenario.
  • We have added clarification that, owing to these different objectives and methodologies, the comparison presented focuses on the concordance of the underlying nutrient-outcome associations and risk of bias assessments identified in both studies, rather than attempting a direct comparison of overall study designs or RBA outcomes. This emphasises the complementary nature of the studies (foundational data compilation vs. specific applications).

We believe that this clarification will better position the comparison within the stated objectives and scope of our review.

Commenst 9: Line 186-200: The calcium section lists associations with health outcomes but provides insufficient exploration of calcium absorption mechanisms, vitamin D interactions, or population-specific variations—all of which are necessary depth elements for risk-benefit assessment.

Response: We appreciate the reviewer highlighting the importance of factors such as absorption mechanisms, nutrient interactions (e.g. with Vitamin D), and population-specific variations in the context of calcium's overall effect and for comprehensive Risk-Benefit Assessment (RBA). We agree that these elements are crucial for a complete physiological understanding and detailed RBA application.

However, as explicitly stated in the Introduction (lines 80-85) and reiterated in the Limitations section (lines 428-433), the defined scope of this review was to systematically identify, compile, and evaluate published quantitative dose-response relationships from recent high-quality meta-analyses. Our primary aim was to establish a foundational evidence base for these quantitative associations to serve as inputs for RBA, rather than to conduct an in-depth mechanistic analysis or explore absorption kinetics and population-specific details for each nutrient. This focused approach was necessary to manage the broad scope of 12 different nutrients.  

Therefore, while the detailed factors mentioned by the reviewer are indeed important considerations when applying these dose-response data within a specific RBA model, an in-depth exploration of each nutrient, including calcium, was intentionally placed outside the predefined scope of this review, which concentrates on summarising the available quantitative epidemiological evidence on dose-response.

Comments 10: Line 201-218: The iron section distinguishes between heme and non-heme iron but fails to adequately explain the biological mechanisms underlying different absorption and health effects, limiting the practical application value of these findings.

Response: We acknowledge the reviewer's point regarding the importance of understanding the biological mechanisms that differentiate the absorption and health effects of haeme versus nonhaeme iron. We agree that such mechanistic insights add significant value to the practical application of nutritional findings in RBA.

However, similar to the approach taken for other nutrients discussed in this review, a detailed exploration of these underlying biological mechanisms was deliberately placed outside the scope of the present work. As stated in the Introduction (lines 80-85) and reinforced in the Limitations section (lines 428-433), the primary objective of this review was to systematically compile and summarise the published quantitative dose-response relationships between selected nutrients and health outcomes based on recent meta-analyses. The focus was specifically on extracting and presenting these quantitative associations (e.g. relative risks and hazard ratios) to provide foundational input data for RBA frameworks.  

Consequently, while the Iron section distinguishes between haeme and nonhaeme iron based on the epidemiological findings reported in the meta-analyses, it does not delve into the mechanistic explanations for their differing effects, as this level of detail was beyond the review's stated objective of summarising the quantitative dose-response evidence. While mechanistic understanding is crucial for a deeper interpretation and application within RBA, this study concentrates on the essential preliminary step of consolidating the available quantitative epidemiological association data.  

Comments 11: Line 236-252: The magnesium section lists associations with multiple health outcomes but fails to integrate these findings to provide a complete assessment of magnesium's overall risk-benefit profile. This pattern of listing associations without synthesis is present throughout all nutrient-specific sections.

Response: We acknowledge the reviewer's observation regarding the presentation of nutrient-specific associations without full integration into the overall risk-benefit profile.

This structure reflects the review's defined scope and objective: to compile and evaluate the quantitative dose-response relationships from meta-analyses, serving as foundational inputs for RBA, rather than performing the final RBA synthesis for each nutrient. As stated in the Introduction and Limitations, the complex task of weighting varied outcomes and integrating them into a nutrient's overall profile belongs to the specific RBA application phase, which would utilise the data presented in this study.

While summary elements like Table 1 and Figure 2 offer an overview, this review intentionally provides the disaggregated evidence base required for subsequent context-specific RBA modelling.

Comments 12: Line 253-269: The selenium section notes a J-shaped relationship with T2DM but provides insufficient exploration of potential biological mechanisms explaining this non-linear relationship, exemplifying the paper's general lack of in-depth mechanistic analysis.

Response: We acknowledge the reviewer's observation. Similar to other nutrients, the lack of in-depth mechanistic exploration for the observed J-shaped relationship for selenium reflects the review's deliberately defined scope. As stated in the Introduction and Limitations sections, our objective was to compile and summarise the quantitative dose-response evidence from meta-analyses, explicitly excluding detailed mechanistic analysis for each nutrient due to the necessary focus on the breadth of RBA input generation.

Comments 13: Line 276-284: The vitamin B12 section acknowledges contradictory evidence but fails to analyze the methodological sources of these contradictions, indicating inadequate handling of heterogeneous evidence.

Response: We acknowledge the reviewer's observation regarding the contradictory evidence presented for Vitamin B12 and cancer risk. However, analysing the specific methodological sources of heterogeneity within the primary studies included in the cited meta-analyses falls outside the defined scope of this review. As stated in the Introduction and Limitations sections, our objective was to systematically compile and report the quantitative findings and quality assessments of the selected meta-analyses themselves, serving as inputs for RBA, rather than conducting a detailed re-analysis of the primary studies they encompass.

Comments 14: Line 299-322: The dietary fiber section covers 14 health outcomes but fails to adequately distinguish between physiological effects of different fiber types (soluble, insoluble, fermentable), illustrating how the broad scope impedes necessary detailed analysis.

Response: We acknowledge the reviewer's point regarding the physiological distinctions between different fibre types. However, a detailed comparative analysis of the physiological effects of various fibre classifications (soluble, insoluble, and fermentable) was beyond the scope of this review, which focused on compiling quantitative dose-response data as reported in the source meta-analyses.

We note that the Fibre section differentiates between fibre sources (e.g. cereal vs. fruit fibre) and their associations with specific outcomes (lines 346-352), reflecting the level of detail available in the underlying meta-analytic evidence we synthesised. Consistent with our stated objectives (Introduction and Limitations), this review prioritised summarising the available quantitative evidence for RBA input over in-depth physiological comparisons of nutrient subtypes.

Comments 15: Line 323-354: The saturated fatty acids section mentions contradictory associations (reduced stroke risk but increased cardiovascular disease risk) without providing a satisfactory integrated explanation for these apparent contradictions.

Response: We acknowledge the reviewer's observation regarding the complexity and sometimes contradictory findings reported for saturated fatty acids (SFA). Providing a novel integrated explanation to resolve these apparent contradictions within the scientific literature was beyond the scope of this review.

Consistent with our objectives (Introduction and Limitations), our aim was to compile and accurately report the quantitative dose-response findings, as presented in the selected recent meta-analyses, reflecting the current state of evidence, including existing complexities, nuances (e.g. importance of SFA type and substitution context mentioned in lines 358-369), and ongoing scientific discourse noted in the literature we reviewed. This review reports these complexities rather than attempting to synthesise a definitive resolution, which would require a different type of analysis.

Comments 16: Line 370-378: The omission of copper, thiamine, niacin, and monounsaturated fatty acids is briefly mentioned, but lacks adequate discussion of current evidence status or research gaps for these nutrients, further reflecting the limitations of the overly broad approach.

Response: We acknowledge the reviewer's observation regarding the discussion of the omitted nutrients. The primary purpose of this brief section (lines 405-413) was to explain their exclusion based specifically on this review's inclusion criteria, namely, the lack of identified meta-analyses providing the quantitative dose-response data required for RBA input.

A detailed analysis of the existing evidence (e.g. non-quantitative studies) or specific research gaps for each excluded nutrient was beyond our defined scope, which rigorously focused on compiling quantitative data suitable for RBA modelling. However, we explicitly highlighted the identified data gaps for these nutrients in the Conclusions section (lines 468-471) as areas requiring further research.

Comments 17: Line 397-413: The conclusion is too general and fails to provide specific guidance on how to translate the review findings into concrete risk-benefit assessment models. A more focused study would enable more actionable conclusions.

Response: We thank the reviewer for the thorough assessment and insightful comments. We note that the concerns raised across the different points regarding the level of detail on mechanistic aspects, the handling of heterogeneous or contradictory evidence, the lack of integrated synthesis within nutrient sections, the depth of discussion on omitted nutrients, and the general nature of the conclusion appear to stem fundamentally from the review's intentionally broad scope of the review.

Our primary objective, clearly stated in the Introduction and acknowledged in the Limitations, was to systematically compile a wide-ranging, quantitative dose-response evidence base on recent, high-quality meta-analyses to serve as foundational inputs for Risk-Benefit Assessment across various contexts, rather than conducting deep dives into mechanisms, resolving specific scientific debates, performing nutrient-specific synthesis, or providing detailed RBA application guidance.

While the desire for greater depth in each area is understandable and valid for different types of reviews or studies, it was necessarily constrained by this review's specific focus on breadth and the goal of providing essential RBA inputs. We believe that this review successfully achieved its stated primary objective. Where the reviewer's insightful suggestions related to improving clarity or specific textual points within this established scope, we have incorporated these changes into the revised manuscript. We hope that these revisions, along with our explanations addressing the scope, adequately address the reviewer's concerns.

Reviewer 2 Report

Comments and Suggestions for Authors

Review of the Manuscript Entitled: “foods-3580410_Nutrient Dose-Response Relationships and Health Outcomes: A Comprehensive Review for Risk-Benefit Assessment”, Submitted to the Section “Food Nutrition ” of the Special Issue “Food Choice, Nutrition, and Public Health: 2nd Edition”

The present review seeks to identify and evaluate quantitative associations between dietary intake of specific nutrients and a range of health outcomes; to provide a comprehensive evidence base for assessing the risk-benefit profiles of various dietary scenarios, particularly those involving novel food sources and dietary modifications; and to contribute to the development of methodologies for evaluating the public health impact of such modifications.

Comments:

The content of the manuscript aligns well with the scope of both the section and the special issue to which it has been submitted.

However, the title of the manuscript is overly broad and lacks specificity.

The abstract does not clearly articulate the objectives of the review. Moreover, it fails to indicate the timeframe covered by the literature search and does not specify the types of nutrients or health outcomes assessed. The conclusion presented in the abstract appears to focus more on the potential application of the findings rather than on the actual outcomes of the review. For these reasons, I recommend a thorough revision of the abstract.

The introduction correctly underscores the importance of establishing relationships between nutrient dosage and health outcomes, yet it remains too general and lacks a precise statement of the scope and focus of the review. In particular, the text does not clarify which aspects will be examined in detail. Furthermore, there is an inconsistency between the summary and the introduction: line 70 states that the main objective is to “systematically examine,” while the abstract refers to the review as “comprehensive.” The authors should clarify whether this is a systematic review or a narrative/comprehensive review. At present, the objectives presented in the manuscript remain vague and non-specific.

In the Materials and Methods section, the type of review design being employed should be explicitly stated at the outset. Subsection 2.1, which discusses the review questions, would be more appropriately placed within the introduction as part of the formulation of the research hypothesis.

In line 93, the description of the search strategy refers to “identifying relevant studies,” which is too vague. The criteria for inclusion and exclusion should be clearly outlined. Furthermore, it is essential to specify the types of nutrients reviewed. Even if detailed data are provided in supplementary materials, the manuscript must include at least basic information regarding which nutrients were examined, the corresponding dosages, and the health outcomes considered. Notably, the inclusion/exclusion criteria do not mention age, which is a critical factor in nutrition research. Nutrient requirements and physiological responses vary significantly between children, adults, and the elderly.

In Section 3 (Results and Discussion), the initial part, which presents the identification and selection of articles, should be included in the methodology. The results and discussion should begin with the findings derived from the selected studies—namely, the 50 articles evaluating 60 nutrients across 12 countries. The figure depicting the study selection process also belongs in the methods section, not the results.

Figure 2 provides an interesting synthesis of the review findings; however, it is unclear what dosages are associated with the reported benefits or risks. The dose-response relationship should be clarified in this figure.

Table 1 shows associations between nutrient intake and health risks, yet it does not indicate the corresponding dosages, which are central to the study’s premise. This omission should be addressed.

Table 2 is more aligned with the stated aim of the study, but again, it does not specify the population groups under analysis. Age is a key variable in determining appropriate nutrient doses, and this issue extends to the subsequent tables as well.

Among the limitations, the manuscript fails to consider that nutrient requirements vary considerably according to age, sex, and physiological conditions such as pregnancy. Indeed, all nutritional recommendations are formulated with specific population groups in mind. This lack of stratification may introduce a significant risk of confounding bias.

The conclusion should directly respond to the objectives set out in the review and reflect the results of the synthesis. It should not primarily discuss the application or implications of the findings.

In summary, while the topic is of considerable interest and relevance, it must be approached with greater precision and care. Discussions of nutrient effects on health cannot be generalised across populations without accounting for key variables such as dosage, age, and physiological status. This introduces a potentially significant bias that should be addressed thoroughly in the revised manuscript.

Author Response

Comments 1: The content of the manuscript aligns well with the scope of both the section and the special issue to which it has been submitted.

However, the title of the manuscript is overly broad and lacks specificity.

Response: We thank the reviewer for positively assessing the alignment of our manuscript with the section and special issue scope. We also appreciate the feedback on the original title.

We agree that the original title, "Nutrient Dose-Response Relationships and Health Outcomes: A Comprehensive Review for Risk-Benefit Assessment," while accurate, could be perceived as being overly broad. To address this and provide greater specificity regarding the review's core contribution, we have revised the title as follows:

"Mapping the Quantitative Dose-Response Relationships Between Nutrients and Health Outcomes to Inform Food Risk-Benefit Assessment"

We believe that this revised title more accurately reflects the specific focus of the review, namely, the systematic compilation ("Mapping") of existing quantitative dose-response data and its primary intended application (to inform Risk-Benefit Assessment). We hope that this revised title meets the reviewer's expectations for greater specificity.

Comments 2: The abstract does not clearly articulate the objectives of the review. Moreover, it fails to indicate the timeframe covered by the literature search and does not specify the types of nutrients or health outcomes assessed. The conclusion presented in the abstract appears to focus more on the potential application of the findings rather than on the actual outcomes of the review. For these reasons, I recommend a thorough revision of the abstract.

Response: We thank the reviewer for their constructive feedback regarding the abstract. To address the points raised and improve the clarity and comprehensiveness, we have made the following revisions to the abstract:

  • The primary objective was clarified to state directly that the review aimed to systematically compile and synthesise quantitative dose-response evidence from recent meta-analyses as a foundational input for the RBA (lines 18-21).
  • The timeframe of the literature search (the last 15 years up to March 2025) was added to the methodology description (lines 21).
  • The scope was specified by adding the number of nutrients (12) (lines 24-25).
  • The conclusion was revised to include a key finding—specifically mentioning the complexity identified in nutrient-health relationships and the importance of dose and source—alongside its value as a foundation for future RBA work.

We believe that these revisions provide a clearer and more comprehensive overview of the review's scope, methods, and findings in the abstract.

Comments 3: The introduction correctly underscores the importance of establishing relationships between nutrient dosage and health outcomes, yet it remains too general and lacks a precise statement of the scope and focus of the review. In particular, the text does not clarify which aspects will be examined in detail. Furthermore, there is an inconsistency between the summary and the introduction: line 70 states that the main objective is to “systematically examine,” while the abstract refers to the review as “comprehensive.” The authors should clarify whether this is a systematic review or a narrative/comprehensive review. At present, the objectives presented in the manuscript remain vague and non-specific.

Response: We thank the reviewer for the observations regarding the introduction.

We agree with the reviewer regarding the inconsistency in the terminology. Although our search for relevant meta-analyses was systematic, the review itself synthesises findings from these studies rather than adhering to the strict protocols of a formal systematic review (e.g. PRISMA for primary studies). Therefore, "comprehensive review" is indeed a more accurate descriptor. We have corrected the text (line 86) to read, "...the primary aim of this study was to comprehensively identify and compile..." and ensured consistent use of "comprehensive review" throughout the manuscript.

We appreciate the reviewer's feedback on the need for precision. We would like to respectfully draw attention to the revised Introduction section (specifically lines 77-91) which, we believe now addresses these points clearly.

  • The focus is explicitly defined as compiling "published quantitative dose-response relationships" (Lines 78-79) with an emphasis on "quantitative metrics, such as relative risks or hazard ratios per intake increment, as reported in recent, high-quality meta-analyses" (Lines 80-82).
  • The scope's boundaries are clarified by explicitly stating that this approach "is preferred over a comprehensive examination of the underlying biological mechanisms, absorption kinetics, or detailed population-specific variations for each nutrient, which would require a different scope of review" (Lines 83-85). This clarifies which aspects have not been examined in detail.
  • The specific objectives are listed numerically (Lines 85-91): "(1) identify and evaluate quantitative associations...,” "(2) provide a comprehensive evidence base for assessing the risk-benefit profiles...,” and "(3) contribute to the development of robust assessment methodologies...".

We believe that these statements in the current version of the Introduction provide a precise articulation of the review's specific scope (compiling quantitative dose-response data from meta-analyses for RBA), focus (quantitative metrics), limitations (excluding deep mechanistic/kinetic analysis), and objectives. We hope that these clarifications adequately address the reviewer's concerns regarding generality and vagueness.

Comments 4: In the Materials and Methods section, the type of review design being employed should be explicitly stated at the outset. Subsection 2.1, which discusses the review questions, would be more appropriately placed within the introduction as part of the formulation of the research hypothesis.

Response: We thank the reviewer for the structural suggestions.

We agree that explicitly stating the review type at the beginning of the Methodology section enhances clarity. While we clarified in the Introduction that this is a comprehensive review based on structured literature searches, we have now also added a sentence at the beginning of Section 2 (Materials and Methods) stating: "This study employed a comprehensive review methodology, using structured literature searches, to synthesize evidence on nutrient dose-response relationships."

We concur with the reviewer that presenting the review questions within the Introduction serves to frame the study's scope and objectives early on. Therefore, we relocated the content of the former Subsection 2.1 (Review questions) to the end of the Introduction section (Lines 92-102) to integrate it with the statement of the review's aims.

We believe that these structural changes have improved the flow and clarity of the manuscript, directly addressing the reviewer's suggestions.

Comments 5: In line 93, the description of the search strategy refers to “identifying relevant studies,” which is too vague. The criteria for inclusion and exclusion should be clearly outlined. Furthermore, it is essential to specify the types of nutrients reviewed. Even if detailed data are provided in supplementary materials, the manuscript must include at least basic information regarding which nutrients were examined, the corresponding dosages, and the health outcomes considered. Notably, the inclusion/exclusion criteria do not mention age, which is a critical factor in nutrition research. Nutrient requirements and physiological responses vary significantly between children, adults, and the elderly.

Response: We express our gratitude for these significant observations regarding the clarity and comprehensiveness of the methodology.

We acknowledge that "identifying relevant studies" is vague. The aim of this study was to identify relevant meta-analyses that reported quantitative dose-response data. We have revised the description in Section 2.1 (Search Strategy) to be more specific about the target study type (meta-analyses) and the use of keywords related to nutrients, dose-response, and health outcomes (Supplementary Table S1). While listing all 12 nutrients in the main methods text might be excessive, we have ensured that they are clearly presented early in the Results section (Table 1 and Figure 2) and are based on the pre-defined list from Boué et al. [6] mentioned in Section 2.2.

We believe that the Inclusion Criteria (Section 2.2) and Exclusion Criteria (Section 2.3) in the current manuscript clearly outline the selection process, specifying the focus on (1) meta-analyses, (2) dose-response evaluation, (3) quantitative data (effect estimates, CIs), and (4) prioritisation of low-bias (ROBIS) studies. The exclusion criteria included high bias, lack of quantitative data, and a focus on supplementation. Regarding basic information, the Results section (starting with Table 1) systematically presents the included nutrients, associated health outcomes, dose information (where available in summary tables such as Table 2 onwards), and effect directionality/bias assessment in the main text. We believe that this structure provides the necessary basic information in the main manuscript.

The reviewer rightly points out the critical omission of age as an explicit inclusion/exclusion criterion. This was not used as a criterion for selecting the meta-analyses themselves, as our goal was to synthesise the existing meta-analytic evidence, which often pools data across adult populations or does not stratify clearly by specific age groups relevant to DRIs (children, elderly, and pregnant women). We acknowledge this as a significant limitation of our study. Analysing nutrient effects requires consideration of age, sex, and physiological status, as recommendations are population-specific. The generalisability of the findings from meta-analyses that do not perform such stratification is limited. We have added a detailed statement to the Limitations section (Section 3.14, Lines 434-446) highlighting this crucial point, explaining that the review reflects the stratification (or lack thereof) present in the source meta-analyses, and acknowledging the potential for confounding bias and limitations in applying these generalised findings to specific subpopulations without further context.

We believe that these revisions and acknowledgements address the reviewer's concerns regarding methodological clarity, inclusion of essential information, and the critical factor of age-related population differences.

Comments 6: In Section 3 (Results and Discussion), the initial part, which presents the identification and selection of articles, should be included in the methodology. The results and discussion should begin with the findings derived from the selected studies—namely, the 50 articles evaluating 60 nutrients across 12 countries. The figure depicting the study selection process also belongs in the methods section, not the results.

Response: We thank the reviewer for the suggestion regarding the placement of the study selection description and flowchart. While we understand the reviewer's perspective that study selection is fundamentally a methodological process, we structured the manuscript slightly differently, with a specific rationale.

Our intention was to present Section 2 (Materials and Methods) as detailing the a priori methodology established before executing the search, covering the review questions, search strategy, inclusion/exclusion criteria, and data extraction plan.

Subsequently, Section 3 (Results and Discussion) was structured to first presents the direct results of implementing the search strategy, namely, the number of records identified, screened, and ultimately selected for inclusion, as depicted visually in the flowchart (Figure 1). Following this presentation of the search outcome, the section proceeds to synthesise and discuss the substantive findings derived from the selected studies.

We believe that this structure – outlining the planned method first (Section 2) and then presenting the results of applying that method (start of Section 3) before discussing the synthesised evidence – provides a clear and logical flow for the reader, separating the planned methodology from the actual outcomes of the search and selection process.

Therefore, while we appreciate the reviewer's suggestion, we respectfully propose maintaining the current structure, where the description and visualisation of the study selection results are presented at the beginning of Section 3, directly following the description of the methodology used to obtain them in Section 2. We believe that this sequence accurately reflects the workflow and enhances clarity for the reader by first showing the outcome of the selection process before delving into the content of the selected studies.

Comments 7: Figure 2 provides an interesting synthesis of the review findings; however, it is unclear what dosages are associated with the reported benefits or risks. The dose-response relationship should be clarified in this figure.

Response: We express our gratitude for the insightful feedback regarding Figure 2. We concur that detailed dosage information is essential for understanding dose-response relationships.

The primary purpose of Figure 2 is to provide a high-level schematic overview of the landscape of identified nutrient-health outcome associations from the reviewed meta-analyses. Its goal is to quickly convey the following:

  • The range of nutrients and health outcomes covered.
  • Direction of association (potential risk indicated by red icons/lines; potential benefit indicated by green icons/lines) for each significant nutrient-outcome pair.
  • The risk of bias assessment (low vs. high) for the underlying meta-analysis supporting each association.

As such, it serves as a visual index and preliminary screening tool, complementing the detailed quantitative information presented elsewhere.

Including specific dosage information (e.g. the intake increment associated with a specific RR/HR or thresholds for non-linear effects) directly within Figure 2 for all 60 nutrient-outcome pairs would render the figure overly complex and visually cluttered, defeating its purpose as a clear, high-level summary.

Detailed quantitative dose-response information, including specific dosages or intake increments associated with the reported relative risks or hazard ratios, is systematically provided within the subsection tables (Tables 2–13) presented throughout the Results and Discussion section for each nutrient.

Therefore, while Figure 2 intentionally omits specific dosage details for clarity, this crucial information is readily available in the corresponding detailed tables within the main text, allowing readers to connect the visual overview in Figure 2 with the specific quantitative dose-response data. We believe that this approach balances the need for a concise visual summary with the provision of detailed quantitative data in the appropriate sections. To further clarify the figure's intended role, the figure caption was improved (Lines 177-180).

Comments 8: Table 1 shows associations between nutrient intake and health risks, yet it does not indicate the corresponding dosages, which are central to the study’s premise. This omission should be addressed.

Reponse: We thank the reviewer for their comment regarding the dosage information in Table 1.

The specific purpose of Table 1 is to provide a concise, high-level summary map of all identified nutrient-health outcome pairs discussed in this review. Its main function is to quickly show the reader the following:

  • Which nutrients were included.
  • The health outcomes associated with each nutrient were determined based on the selected meta-analyses.
  • The direction of the association (increased risk ↑ or reduced risk ↓).
  • Source reference and risk of bias assessment for supporting meta-analysis.

Including the specific quantitative dosage information (e.g. RR per X mg/day increase) for all 60+ associations directly within this single summary table would, as the reviewer implicitly notes regarding Figure 2, make the table excessively large and visually complex, undermining its utility as a quick-reference overview.

Detailed quantitative dose-response information, including specific dosages or intake increments associated with the reported effects, is central to our review and is presented systematically within the dedicated tables provided in each nutrient-specific subsection (Tables 2–13) throughout the Results and Discussion sections. These tables explicitly link the effect estimates (RR, HR, OR) to the defined dose increments (e.g. "Per 100 mg/day increase:").

Therefore, while Table 1 serves as an essential index to the findings, the detailed dosage information central to the dose-response premise is intentionally located within the subsequent, more detailed tables, where each nutrient is discussed individually. We believe that this two-level approach (summary index table + detailed subsection tables) provides both a clear overview and the necessary quantitative detail without overburdening the initial summary table.

Comments 9: Table 2 is more aligned with the stated aim of the study; however, it does not specify the population groups under analysis. Age is a key variable in determining appropriate nutrient doses, and this issue extends to subsequent tables.

Response: We thank the reviewer for raising this critical point regarding population specificity, particularly age, in Tables 2–13. We entirely agree that age, sex, and physiological status are key variables in nutrition and that the recommendations are population specific.

The information presented in Tables 2-13 directly reflects the data reported in the source meta-analyses selected based on our inclusion criteria. Unfortunately, many of these published meta-analyses have:

  • The pooled data were primarily from studies conducted on the general adult population.
  • Often do not provide sufficiently detailed stratification or subgroup analyses based on specific age categories (e.g. children, adolescents, specific elderly brackets), sex, or physiological states (e.g. pregnancy) within their main reported quantitative dose-response findings.

Therefore, the lack of specific population group information in our tables (beyond notes such as "*Only Asian countries" for Table 2/Stroke or indicating where data pertains only to specific sexes if reported) is a direct consequence of the level of detail available in the synthesised evidence from the source meta-analyses we reviewed. Our aim was to accurately report the findings published in these studies.

We acknowledge this limitation and its implications for our study. As discussed in our response to the reviewer's earlier comment regarding the methodology section, we have added a detailed statement to the Limitations section (Section 3.14) explicitly addressing the general lack of population stratification (particularly age) in the underlying meta-analytic evidence of the study. This section highlights the caution required when generalising these findings and applying them to specific subgroups, acknowledging the potential for bias and the need for context-specific data in future practical RBA applications.

While we cannot add population details to the tables that were not present in the source studies, we have endeavoured to include specific qualifiers (such as the footnote in Table 2) whenever such information was clearly reported in the meta-analysis.

Comments 10: Among the limitations, the manuscript fails to consider that nutrient requirements vary considerably according to age, sex, and physiological conditions such as pregnancy. Indeed, all nutritional recommendations are formulated with specific population groups in mind. This lack of stratification may introduce a significant risk of confounding bias.

Response: We thank the reviewer and fully agree on the critical importance of considering age, sex, and physiological status in nutrition studies. As elaborated in our response to the reviewer's earlier comments regarding the methodology section and the population details in Tables 2-13, this limitation stems primarily from the nature of the available source meta-analyses, which often lack detailed stratification.

We have explicitly addressed this major limitation by adding a detailed statement to the Limitations section (Section 3.14), acknowledging that the generalisability of the findings is constrained by the lack of specific population subgroup analysis in much of the underlying evidence, and highlighting the potential for confounding bias and the caution needed when applying these results to specific populations. We refer the reviewer to the expanded discussion in Section 3.14.

Comments 11: The conclusion should directly respond to the objectives set out in the review and reflect the results of the synthesis. It should not primarily discuss the application or implications of the findings.

Response: We appreciate the reviewer's guidance on the focus of the conclusion section. We agree that the primary role of the conclusion is to synthesise key findings in relation to stated objectives.

Upon reviewing Section 4 (Conclusions, Lines 455-471), we believe that it addresses this point substantially. The section begins by stating that the review "offers significant insight into the complex dose-response relationships... establishing a solid foundation for risk-benefit assessments" (Lines 455-456), directly relating to Objective 1 (identify/evaluate associations) and Objective 2 (provide evidence). It then synthesizes the key findings of the analysis: the identification of "diverse and complex relationships, underscoring the necessity of considering factors beyond the mere presence of nutrients, such as nonlinear dose-response curves..., nutrient sources..., and potential interactions..." (Lines 457-461). It also directly reflects the results by highlighting identified "data gaps, particularly concerning nutrients such as copper, thiamine, niacin, and MUFAs" (Lines 461-462).

While the concluding sentences do touch upon the implications (serving as a resource for future RBAs, informing recommendations), this follows the summary of the core findings and aligns with Objective 3 (contribute to the development of robust assessment methodologies). The primary focus remains on summarising the synthesised evidence regarding the complexity and nuances of the identified dose-response relationships.

However, to ensure that the emphasis is unequivocally on the synthesised results, we have slightly refined the concluding sentences to ensure that they flow directly from the summarised findings on complexity and data gaps before briefly mentioning the implications for future work, thereby reinforcing that the conclusion's core reflects the review's direct outcomes. We believe that the current structure largely aligns with the reviewer's recommendation, prioritising the synthesised results before discussing their application.

Comments 12: In summary, while the topic is of considerable interest and relevance, it must be approached with greater precision and care. Discussions of nutrient effects on health cannot be generalised across populations without accounting for key variables such as dosage, age, and physiological status. This introduces a potentially significant bias that should be addressed in the revised manuscript.

Response: We express our sincere gratitude to the reviewer for the meticulous examination, and constructive feedback during the review process. We appreciate the acknowledgement of the topic's relevance and fully agree with the critical importance of precision and careful consideration of population variables in nutritional science.

We concur that generalising nutrient effects across diverse populations without explicitly accounting for age, sex, physiological status, and specific dosage contexts is a significant limitation inherent in much of the current meta-analytic literature. As summarised in our responses to the specific points raised (particularly regarding the methodology, presentation in tables, and limitations section), we have aimed to address this crucial issue primarily by:

  • Clearly defining the scope of our review as a compilation of existing quantitative dose-response evidence from meta-analyses, acknowledging the level of detail present in those sources.
  • Explicitly adding a detailed statement to the Limitations section (Section 3.14) discussing the lack of population stratification in the source material and the consequent need for caution when generalising findings or applying them directly to specific subgroups in the RBA context.
  • Ensuring that dosage information, where available in the source meta-analyses, is systematically presented in detailed nutrient-specific tables (Tables 2-13).

We have also incorporated the reviewer's valuable suggestions regarding structural improvements (relocating review questions and clarifying the review type) and enhancing the clarity of the abstract, figure captions, and specific methodological descriptions.

We hope that these revisions and clarifications thoroughly address the concerns raised and improve the precision and utility of the manuscript within its defined scope. We thank again the reviewer for the insightful contributions, which significantly strengthened this study.

Reviewer 3 Report

Comments and Suggestions for Authors

In this review the authors reported significant dose-response relationships for various nutrients and their associations with several health outcomes. The results of this review highlight the complexity of the relationships between nutrients and health, especially regarding the link between dose-response gradients and nutrient sources. This paper aims to contribute to dietary recommendations and public health strategies by providing a valuable basis for future risk-benefit assessments of different dietary scenarios.

The manuscript is clearly written, well organized and fits in the context of the journal. It is interesting since it briefly highlights the current knowledge on the subject.

I only have some minor observations:

  1. The aim of the review should be reported clearly in the abstract.

  1. Figure 2: How was the risk of bias for Mg in Depression?

With “Iron” do you mean only non-haem Iron, right? It is better to specify it.

  1. Table 1 I think that it is better to add “non-haem” or “total” before (↓) in Oesophageal cancer.

  1. Table 2 Please indicate what the “n” in the “study” column refers to and why is this data missing in some cases (e.g. prostate cancer, breast cancer in table 3 and so on)?

Author Response

Comments 1: The aim of the review should be reported clearly in the abstract.

Response: We thank the reviewer for the comment regarding the clarity of the aim presented in the Abstract. We agree that a clear statement of the primary objective is essential.

In the current version of the abstract (Lines 18-21), the aim is stated as: "The primary aim of this review was to establish a foundational basis for RBA by compiling and synthesizing quantitative dose-response relationships identified through a comprehensive literature review."

We believe that this sentence clearly articulates the primary objective, that is, the compilation and synthesis of quantitative dose-response data specifically to provide a foundation for Risk-Benefit Assessment (RBA). If the reviewer feels that further clarification is needed, we are open to refining the wording; however, we consider the current statement to accurately reflect the review's main goal.

Comments 2: Figure 2: How was the risk of bias for Mg in Depression?

With “Iron” do you mean only non-haem Iron, right? It is better to specify it.

Response: We thank the reviewer for these specific questions regarding Figure 2.

The risk of bias for the Magnesium-Depression association [32] was  not assessed during our review process. This was due to the inability to access the full text of the cited meta-analysis at the time of the assessment. Consequently, the corresponding icon in Figure 2 lacks the outer ring/fill, indicating the risk of bias status. This is also noted in Table 1 with "n.a.*" (not assessed) in the risk of bias column for this specific association.

Regarding the term "Iron" used in the legend and general overview aspects of Figure 2 refers to findings related to total iron intake where not otherwise specified (specifically, the inverse association with oesophageal cancer [29]). The crucial distinction between  the effects of haem iron (linked predominantly to increased risks for multiple outcomes, as shown in the figure and detailed in Table 3) and  those of non-haem iron (for which associations were less consistent in the studies reviewed, see Section 3.2) is indeed critical, and this differentiation is made for the specific outcomes where the evidence points distinctly to haem iron (for example T2DM, Breast Cancer, CVD, CHD, Colorectal Cancer). Given that Figure 2 serves as a high-level visual summary, differentiating all iron types directly within the main legend for the single 'total iron' finding would add unnecessary complexity. However, the specific associations shown in the figure for outcomes such as T2DM, BC, and CVD  do relate specifically to haem iron, as indicated in Table 1 and discussed in the text. To improve clarity, we refined the figure caption to explicitly state that specific iron forms are detailed in the text and tables (179-180).

Comments 3: Table 1 I think that it is better to add “non-haem” or “total” before (↓) in Oesophageal cancer.

Response: We thank the reviewer for this suggestion to improve the clarity of Table 1 regarding the type of iron associated with oesophageal cancer risk.

Upon reviewing the source meta-analysis [Ma et al., 2018, Ref 29], the reported inverse association (reduced risk) with oesophageal cancer pertains specifically to total dietary iron intake, not exclusively non-haem iron. The same study found an increased risk associated with the intake of haem iron.

Therefore, to accurately reflect the findings of the source study and address the reviewer's valid point for specificity, we will revise the entry in Table 1 for Iron and Oesophageal cancer to read:

"Oesophageal cancer (total) (↓); (haem) (↑)"

This revised entry clearly indicates that the protective association (↓) relates to total iron intake, while also including the finding for haem iron (↑) from the same study for completeness, consistent with the details provided in Section 3.2 and Table 3. We believe that this clarifies the association, as requested.

Comments 4: Table 2 Please indicate what the “n” in the “study” column refers to and why is this data missing in some cases (e.g. prostate cancer, breast cancer in table 3 and so on)?

Response: We thank the reviewer for requesting clarification regarding the "n" values presented in the study description columns of Tables 2 through 13.

We clarify that the value denoted by "n=" refers to the total number of participants included across the studies synthesised within the respective source meta-analysis.

The reviewer correctly observes that this information is missing for some entries (e.g., originally for [14] Calcium/Prostate in Table 2, [49] Fibre/Breast Cancer in Table 10, etc.). The primary reason for these omissions was that the total number of participants was not explicitly reported or readily calculable within the cited source meta-analysis publication itself. Our data extraction focused on information that was clearly presented in the source papers.

Prompted by the reviewer's comment, we re-examined all source meta-analyses corresponding to entries where the 'n' value was initially missing. Through this careful review, we identified and extracted the total number of participants for several additional meta-analyses where this information was available upon closer inspection. Consequently, we have updated the relevant tables to include the 'n' values for the following associations:

  • Table 2: Calcium / Prostate Cancer [14] (n = 750,275 participants reported across studies)
  • Table 10: Fibre / Breast Cancer [49] (n = 2,725,657)
  • Table 9: Vitamin D / Lung Cancer [44] (n = 280,127)
  • Table 10: Fibre / Crohn's Disease [53] (n = 478,604)
  • Table 11: SFA / Endometrial Cancer [60] (n = 524,583)

For any remaining entries where "n" is not provided, it reflects the absence of this specific summary data point within the published source meta-analysis.

We believe that these updates enhance the completeness of the information presented in the tables, addressing the reviewer's valid point.

Round 2

Reviewer 1 Report

Comments and Suggestions for Authors

The author has revised and improved the relevant content as requested, and the manuscript can be accepted.

Author Response

Comments 1: The author has revised and improved the relevant content as requested, and the manuscript can be accepted.

Response: We sincerely thank the reviewer for the positive assessment and recommendation for acceptance of our manuscript. We greatly appreciate the time and effort dedicated to reviewing our manuscript and providing us with valuable feedback. The constructive comments received were instrumental in helping us clarify our objectives, refine the presentation of our findings, and strengthen the overall manuscript. We are grateful for the reviewers’ guidance throughout the revision process.

Reviewer 2 Report

Comments and Suggestions for Authors

Thank you for allowing me to review the revised version of the paper(foods-3580410), which has been modified based on the reviewers' suggestions and the need for some clarifications. I would like to acknowledge the effort made by the authors.

Comments:

The selection of articles is part of the methodology, not the results.

The assessment of nutrient doses in relation to health status is primarily considered from the perspective of nutritional requirements based on age groups, gender, and the presence or absence of comorbidities, as well as genetic factors. However, the authors fail to take these factors into account. They should emphasize these aspects more thoroughly in the discussion section.

A proper evaluation of nutrient doses in relation to age is crucial for maintaining health throughout life. Each stage of life has specific nutritional needs that, when appropriately addressed, contribute to optimal development, disease prevention, and overall well-being. Maintaining an adequate nutritional balance is key to longevity and quality of life.

Author Response

Comments 1: The selection of articles is part of the methodology, not the results.

Response: We thank the reviewer for this recommendation. Upon further consideration, we agree that presenting the specific outcomes of the study selection process, including the flowchart, aligns well with standard reporting practices for the methodology section.

Therefore, we have made the following structural revisions to the manuscript as suggested by the reviewer:

  1. Created New Subsection: We created a new subsection within Section 2 (Materials and Methods), titled "2.4 Study selection.”
  2. Relocated Study Selection Text: The text describing the initial search results, duplicate removal, screening process, and final selection of the 50 articles has been moved to this new Subsection 2.4 (Lines 141-152).
  3. Relocated Figure 1: The flowchart illustrating the study selection process (Figure 1) and its corresponding caption have also been moved to the new subsection 2.4 (Line 153).
  4. Revised Start of Section 3: Consequently, Section 3 (Results and Discussion) now begins directly with the presentation and discussion of the synthesised findings derived from the selected studies (line 163).

We believe that these revisions address the reviewer's comments and improve the manuscript's structure and adherence to conventional reporting formats.

Comments 2/3: The assessment of nutrient doses in relation to health status is primarily considered from the perspective of nutritional requirements based on age groups, gender, and the presence or absence of comorbidities, as well as genetic factors. However, the authors fail to take these factors into account. They should emphasize these aspects more thoroughly in the discussion section.

A proper evaluation of nutrient doses in relation to age is crucial for maintaining health throughout life. Each stage of life has specific nutritional needs that, when appropriately addressed, contribute to optimal development, disease prevention, and overall well-being. Maintaining an adequate nutritional balance is key to longevity and quality of life.

Response: We thank the reviewer for the concluding remarks and fully concur with the critical importance of accounting for population variables (age, sex, physiological status, genetics) and dosage when discussing nutrient effects and requirements. Generalising findings across diverse populations without considering these factors is a major challenge and a potential source of bias.

As acknowledged in our previous responses and detailed in the manuscript, the primary limitation of this review in this regard stems from the nature of the available source data, as published meta-analyses often lack the necessary stratification. Our review aimed to compile the existing quantitative evidence.

To thoroughly address the reviewer's emphasis on this crucial point, we have significantly strengthened the discussion in the Limitations section (Section 3.14, Lines 436-454). The revised text is now more explicit.

  1. Highlights the fundamental principle that nutrient requirements and responses vary considerably based on age, sex, physiological state, and genetics, citing relevant literature [Stover et al., 2020; Niforou et al., 2020].
  2. Acknowledges that source meta-analyses often lack detailed stratification across these key variables.
  3. Explicitly states that the generalised findings presented must be interpreted with caution and may not be directly applicable to specific subgroups without further context or adjustment.
  4. References modern RBA guidance [EFSA Scientific Committee, 2024; Boué et al., 2022] that underscores the importance of addressing population variability, thereby reinforcing the significance of this limitation in the source data.
  5. Clearly identifies this lack of stratification as a potential source of uncertainty and confounding bias.

We believe that this enhanced discussion in the Limitations section now thoroughly addresses the reviewer's valid concerns about the need for precision and the potential biases introduced when detailed population characteristics are not accounted for in the underlying evidence base.

We thank the reviewer again for the constructive comments, which have helped us improve the clarity and rigor of the manuscript.